# Small Molecule Targeting Immune Cells: A Novel Approach for Cancer Treatment

**DOI:** 10.3390/biomedicines11102621

**Published:** 2023-09-24

**Authors:** Shilpi Singh, Debashis Barik, Ananta Prasad Arukha, Sujata Prasad, Iteeshree Mohapatra, Amar Singh, Gatikrushna Singh

**Affiliations:** 1Department of Neurosurgery, University of Minnesota, Minneapolis, MN 55455, USA; 2Center for Computational Natural Science and Bioinformatics, International Institute of Information Technology, Hyderabad 500032, Telangana, India; 3MLM Medical Labs, LLC, Oakdale, MN 55128, USA; 4Department of Veterinary and Biomedical Sciences, University of Minnesota—Twin Cities, Saint Paul, MN 55108, USA; 5Schulze Diabetes Institute, Department of Surgery, University of Minnesota, Minneapolis, MN 55455, USA

**Keywords:** immunotherapy, tumor microenvironment, innate-like cells, killer innate-like T cells, hybrid cells, tissue-resident memory T cells, regulatory T cells, myeloid-derived suppressor cells, small-molecules

## Abstract

Conventional and cancer immunotherapies encompass diverse strategies to address various cancer types and stages. However, combining these approaches often encounters limitations such as non-specific targeting, resistance development, and high toxicity, leading to suboptimal outcomes in many cancers. The tumor microenvironment (TME) is orchestrated by intricate interactions between immune and non-immune cells dictating tumor progression. An innovative avenue in cancer therapy involves leveraging small molecules to influence a spectrum of resistant cell populations within the TME. Recent discoveries have unveiled a phenotypically diverse cohort of innate-like T (ILT) cells and tumor hybrid cells (HCs) exhibiting novel characteristics, including augmented proliferation, migration, resistance to exhaustion, evasion of immunosurveillance, reduced apoptosis, drug resistance, and heightened metastasis frequency. Leveraging small-molecule immunomodulators to target these immune players presents an exciting frontier in developing novel tumor immunotherapies. Moreover, combining small molecule modulators with immunotherapy can synergistically enhance the inhibitory impact on tumor progression by empowering the immune system to meticulously fine-tune responses within the TME, bolstering its capacity to recognize and eliminate cancer cells. This review outlines strategies involving small molecules that modify immune cells within the TME, potentially revolutionizing therapeutic interventions and enhancing the anti-tumor response.

## 1. Introduction

The gradual development of knowledge in tumor biology improves the conventional strategy of cancer treatment. Surgical treatment, radiotherapy, and chemotherapy are the classical approaches utilized to treat cancer. Surgical operation is the first line of treatment for solid tumors used for early stage cancer patients, and in many instances, tumor reoccurrence happens. Radiotherapy and chemotherapy are conventional cytotoxic therapies with multiple side effects, collateral damage, and killing cancer and normal cells. The non-specific effect of these therapies significantly develops cytotoxicity, resistance, and side effects for other metabolic processes that have a severe impact on the quality of a patient’s life [1].

To overcome the accumulative effect of conventional cancer treatment, immunotherapy emerged as the targeted strategy to combat advanced cancers. Broadly, immunotherapy is defined as the therapy that utilizes naturally derived components or synthetically generated compounds to activate the immune system for combating tumor evasiveness and eliciting tumor destruction. Cancer immunotherapies are antibodies that offer benefits in terms of efficacy and selectivity. This includes monoclonal antibodies (mAbs), oncolytic viruses (talimogene laherparepvec or T-VEC), cell-based therapies including adoptive cell therapy, chimeric antigen receptor (CAR-T cell therapy), T cell infusion, cancer-specific vaccines, immune checkpoint inhibitor (ICI) and combination of tumor-specific vaccines with ICIs (Figure 1). These targeted immunotherapies stimulate immune cells against tumor microenvironment (TME) by activating the immune system or blocking immune inhibitory pathways [2].

Cancer immunotherapies have brought significant advancements in cancer treatment. Immunotherapies targeting T cells encounter a broad range of solid and hematologic malignant cancers exhibiting historically poor prognoses and have provided compelling proof-of-concept to reduce patient tumor progression. The introduction of ICIs against programmed cell death protein-1 (PD-1) and cytotoxic lymphocyte antigen-4 (CTLA-4) has demonstrated an operative impact on the immune system to counter tumor cells within the TME [3]. In addition, advanced cancer treatment strategies have evolved as a combination of two or more therapies to enhance the efficacy of the therapeutic agents by targeting multiple pathways in the TME. The combination therapy reduces the possibility of resistance development using conventional monotherapy and works synergistically to reduce the individual dose of a single drug treatment. Combining multiple drugs potentially kills the cancer cells and boosts the immune cells to control the tumor progression. More than 2000 combination therapies have been under clinical trial for cancer treatment (https://clinicaltrials.gov (accessed on 10 June 2023)). Appendix A describes the combination of different therapeutic approaches used for cancer treatment.

The significant challenges of mono or combinational immunotherapy are that all patients do not respond equally to immunotherapy and exhibit resistance or experience immunological adverse effects. In addition, it has pharmacokinetic (PK) restrictions, such as no oral bioavailability, lengthy half-lives, and inadequate penetration into the tumor. The current immunotherapies, including CAR-T and T cell checkpoint inhibitors, display a need for understanding due to patient diversities having either resistance or tumors not responding to the treatment [4]. In addition, a combination of immunotherapy with conventional therapy increases the risk of toxicities and drug interactions. Identifying the optimal drug combination and dosing regimens can also be challenging [5]. Moreover, the heterogeneity of tumors and individual patient responses further complicate treatment outcomes [6,7]. To address the challenges associated with using and combining existing immunotherapies for resistant and relapsed tumors, it is crucial to promptly identify novel approaches that can enhance the effectiveness of immunotherapies across various patient groups.

Small molecule-based advanced cancer immunotherapy has evolved in recent years, producing better efficacy than antibody-based immunotherapy. It offers the opportunity to specifically target various cellular processes and signaling pathways that regulate immune responses. This diversity modulates immune checkpoint proteins and enhances T cell activation and infiltration into tumors, which promotes anti-tumor immune response regulation. Small molecules can target specific molecular pathways within immune cells and make it easy to target the specific subset of TME components, potentially reducing off-target effects and systemic toxicities [8]. Their versatility allows for combination with other therapies, enabling synergistic effects and enhancing anti-tumor activity. Additionally, it blocks the specific receptors of the immune cells in the TME to promote T-cell-mediated tumor elimination by reprogramming the TME and tumor-associated macrophages (TAMs) [9]. Moreover, the small molecule inhibitors reduced the infiltration of myeloid-derived suppressor cells (MDSCs), regulatory T-cells (Tregs), and TAMs [10]. Small molecules present the opportunity to precisely target molecular irregularities within a patient’s tumor, enabling the development of personalized treatment strategies. Additionally, the convenience of oral administration sets small molecules apart from traditional immunotherapies [7]. This review emphasizes the significance of employing personalized small molecules independently or in conjunction with current immune cell-targeting immunotherapies. Such an approach can reshape the TME to impede tumor growth. This method offers a solution to existing limitations by providing targeted interventions, easy administration, and the potential for heightened effectiveness in combination with immunotherapies. Ultimately, this approach could pave the way for recognizing immunomodulatory small molecule inhibitors as an innovative means to counteract tumor immune evasion and enhance the efficiency of immunotherapy.

## 2. Targeted Small Molecule Therapy

Targeted small-molecule cancer therapy has several benefits over conventional and immunotherapy. There has been a massive evolution in the utilization of small molecules for cancer treatment in the past 20 years [8,10]. Many small molecule compounds have been examined in clinical trials and approved for cancer treatment. These small molecules show potential in penetrating the TME, enhancing immune cell accessibility to precisely target cancer or immune cells within the TME. This leads to increased potency and reduces non-specific cytotoxicity, thus advancing personalized medicine and precision oncology. Therefore, targeted small-molecule therapies have been a growing field of interest in cancer immunotherapy [9]. In 2001, FDA approved the first small molecule compound, “imatinib”, a tyrosine kinase inhibitor (TKI), for clinical use in cancer treatment [11]. Recent studies have revealed that therapeutic doses of Imatinib impact diverse immune cell functions, including the differentiation of dendritic cells (DCs), impairing the proper function of T-cells and macrophages [12,13].

Targeted cancer therapy drugs can be grouped into two categories: macromolecules and small molecules, including nucleic acids, polypeptides, monoclonal antibodies, and antibody-drug conjugates. Small molecule targeted therapies have advantages over macromolecule medications, including patient acceptability, PK characteristics, drug transportation, storage, and prices. Small molecule therapies continue to develop significantly compared to the macromolecule represented by monoclonal antibodies. Eighty-nine anti-cancer small molecule inhibitors have been approved in the United States and China for cancer treatment [11]. These inhibitors target kinases, cellular proteins that regulate epigenetic processes, proteasomes, and enzymes that repair DNA damage. Extensive reviews have elaborated on the inhibitors’ properties, mechanistic perspective, and their targeting proteins [11,14,15]. Further, Appendix A summarizes the small molecules recently approved by the FDA for targeting various candidates of essential pathways involved in different cancers.

## 3. Targeting Tumor Microenvironment: A Possible Route for Tumor Containment

The tumor microenvironment is a hypoxic and acidic environment consisting of non-cellular and cellular components that regulate tumor development and progression via angiogenesis [16,17]. The non-cellular components of TME consist of insoluble or soluble elements, such as extracellular matrix, interstitial fluids, metabolites, growth factors, cytokines, and chemokines. On the other hand, the cellular components are broadly consisting of non-immune (cancer-associated fibroblasts, cancer-associated adipocytes, tumor-associated pericytes, tumor endothelial cells, exosomes) and immune cells (natural killer T, tumor-infiltrating NK, invariant NKT, neutrophils, tumor-associated neutrophils, macrophages, myeloid-derived suppressor, dendritic cells, cytotoxic T lymphocytes, regulatory T cells, helper CD4^+^ T, cytotoxic CD8^+^ T and tissue-resident memory T cells) (Figure 2) [18]. During tumor progression, these components of TME rearrange the tumor composition with different genetic and phenotypic heterogeneous cell populations [17]. The remodeled TME influences the prognosis and the effectiveness of the cancer treatment [19] (Table 1). Small molecules targeting the TME components hold promise in developing a genuine cure to impede tumor advancement and progression. Therefore, targeting specific subsets of the TME emerges as a novel strategy for advancing cancer treatment.

### 3.1. Targeting Non-Immune Cells for Cancer Therapy

#### 3.1.1. Cancer-Associated Fibroblasts

Cancer-associated fibroblasts (CAFs) play a crucial role in shaping the tumor microenvironment and influencing immune cells’ behavior, which drives tumor progression. These specialized cells have been identified as significant contributors to the immunosuppressive conditions present within tumors. Specifically, CAFs impede the activity of vital immune cells like T cells and natural killer (NK) cells, essential for identifying and eliminating cancerous cells. CAF-secreted IL-8 facilitated the recruitment of macrophages to the tissue stroma. The IL-8/CXCR2 pathway is responsible for regulating the polarization of monocytes into the M2 phenotype. Neutralization of IL-8 using antibodies or CXCR2 antagonists like danirixin reduced monocyte chemotaxis [20]. Additionally, CAF-secreted CCL2 contributes to monocyte recruitment within the tumor microenvironment [21].

Recent advancements reveal the potential of small molecule inhibitors to target CAFs and disrupt their interactions with immune cells. By selectively inhibiting specific signaling pathways responsible for CAF activation and function, these small molecules offer a promising strategy to counteract CAF-induced immunosuppression [22]. The primary objective of these inhibitors is to reinvigorate the immune response within the TME, thus enhancing the efficacy of immunotherapies. Although the development and application of small molecule inhibitors for CAFs are currently in experimental stages, they represent a hopeful avenue for therapeutic intervention. These small molecule inhibitors could significantly enhance cancer treatments by influencing CAF-related factors contributing to immune evasion and tumor progression, leveraging the immune system’s potential to combat tumors effectively.

#### 3.1.2. Cancer-Associated Adipocytes

Cancer-associated adipocytes (CAAs) significantly influence immune cells, impacting tumor growth and immune cell metabolism. For instance, adipocyte-derived leptin can modify the metabolic profile of CD8^+^ T cells by triggering STAT3-FAO activation and inhibiting glycolysis. This alteration in metabolic pathways results in the suppression of CD8^+^ T cell effector functions [23]. Furthermore, CAAs release various molecules such as monocyte chemoattractant protein-1 (MCP-1), hepatocyte growth factor, plasminogen activator inhibitor-1, cathepsin S, CCL2, TNFα, VEGF, TGF-β, and IL-8. These molecules attract myeloid cells to the tumor microenvironment, hindering their potential to differentiate into M2/MDSC and promoting angiogenesis [24,25]. To counter these effects, the exploration of small molecule inhibitors has gained attention, intending to target CAAs, disrupt their interactions with immune cells, and specifically intervene in signaling pathways as a crucial part of cancer treatment strategies.

#### 3.1.3. Tumor Endothelial Cells

Tumor endothelial cells (TECs) substantially influence immune cells’ behavior within the tumor growth environment. These specialized cells promote angiogenesis and initiate tumor formation [26]. TECs can respond to type I and type II interferons (IFNs) by overexpressing IDO1, which has been linked to suppressing T cell activation and creating an immunosuppressive microenvironment. TECs express and up-regulate IDO1 in response to type I IFN (αβ) and type II IFN (γ) stimulation in the TME, which has been associated with impeded T cell activation, thus promoting an immunosuppressive micro-environment [27]. Studies using genetically modified mouse models of PDGF-induced gliomas have revealed that perivascular nitric oxide, triggering notch signaling, enhances stem-like properties, further highlighting the complex interplay [28,29]. Consequently, exploring small molecule interventions that target these cells holds promise for advancing cancer treatment strategies.

#### 3.1.4. Extracellular Vesicles

Extracellular vesicles (EVs) exert a considerable impact on the behavior of immune cells within the TME. These vesicles play a pivotal role in cell-to-cell communication by transferring various molecules, including proteins, nucleic acids, and lipids, to neighboring or distant cells. In the context of cancer, EVs derived from tumor cells and other components of the TME can modulate immune cell responses and contribute to tumor progression, invasion, and metastasis by transferring oncogenic proteins, mRNA, and miRNA to recipient cells.

EVs released from cancer cells can carry immunosuppressive factors that dampen the activity of immune cells, such as T cells and NK cells, enabling the tumor to evade immune surveillance. Notably, exosomes produced by cancer cells can lead to reduced expression of the NKG2D receptor on NK cells, impairing their cytotoxic functions and furthering immune evasion strategies [30]. Additionally, tumor-derived EVs have demonstrated an impact on CD8^+^ T cells. Hepatocellular carcinoma cell-produced exosomes deliver 14-3-3 ζ to T lymphocytes, resulting in altered cellular pathways and compromised anti-tumor effectiveness [31]. Further studies have revealed the role of tumor exosomes influencing the immune system by binding to the tumor-reactive antibodies; otherwise, these antibodies target the tumor cells to inhibit the antibody-dependent cellular cytotoxicity [32]. Additionally, exosomes secreted by gastric cancer cells trigger TANs’ N2 polarization and support the migration of cancer cells [33]. They can attract CCL20-expressing Tregs into the TME and cause MDSC proliferation by activating STAT-3 via membrane-associated Hsp72 [34]. Exosomes have been found to have a role in transforming CD4^+^ CD25 T cells into CD4^+^CD25^hi^FoxP3^+^ Treg cells, which produce higher levels of perforin, CTLA-4, IL-10, granzyme (Gzm) B, TGF-1β and Fas ligand [35]. Recent investigations have demonstrated that the tumor exosomes activate the NFkB pathway and enhance the expression of pro-inflammatory cytokines, including IL-8 and IL-1β, which boost macrophage polarization into the M2 phenotype [36].

On the other hand, EVs released by immune cells can also influence the tumor or the immune response directed against it. For instance, exosomes produced by NK cells contain elements like GzmA, perforin, GzmB, granulysin, and FasL, which can exert cytotoxic effects on tumor cells [37]. Moreover, recent advancements have demonstrated that exosomes carrying CARs derived from modified CAR-T cells can effectively target tumor cells [38]. Furthermore, exosomes secreted by cytotoxic T lymphocytes (CTLs) can activate bystander CTLs via low-affinity antigens [39]. Nonetheless, it is essential to understand that the intricate interplay between EVs and immune cells holds significant promise for advancing cancer treatment strategies. In addition, utilizing these EVs as cargo to deliver the small molecules to the immune cells will enhance the ability to combat tumors effectively.

### 3.2. Targeting Immune Cells for Cancer Therapy

#### 3.2.1. Tumor-Infiltrating Natural Killer Cells

Tumor-infiltrating natural killer (TINK) cells are a subset of immune cells that can recognize and eliminate cancer cells. These cells play a critical role in the TME, and the factors of TME can activate NK cells to enhance the anti-tumor activity. These TME factors also promote the growth and survival of cancer cells or suppress the immune cell’s activity. Specifically, the TME influences TINK’s behavior and exhibits changes in phenotypic polarization, cytotoxic ability, degranulation, decreased IFN-γ, and increased VEGF expression [40]. Previous studies identified that high CD27 and CD11b NK cells could convert to MDSCs in the TME due to granulocyte-macrophage colony-stimulating factor (GM-CSF [41]. Although NK cells play an important role in removing tumor cells as cytotoxic innate lymphoid cells (ILCs), the subpopulations of ILCs exhibit multiple functions. These cells are abundant in the lymphoid and mucosae tissues. Three categories of non-cytotoxic ILCs include T-bet^+^ ILC1, GATA3^+^ ILC2, and RORγt^+^ ILC3 cells. The T-bet^+^ ILC1 cells release IFN-γ and TNF-α; GATA3^+^ ILC2 cells secreting IL-13, IL-9, IL5, IL-4 and the third category RORγt^+^ ILC3 cells release GM-CSF, IL-17A, and IL 22. CCR6^+^ cells are also included within the third category of ILC3, which secretes GM-CSF, IL-22, IFN-γ, and TNF-α [42]. Interesting trans-differentiation possibilities exist between ILC2 and ILC3 subsets and ILC1 cells. As a result, they can gain or lose specific cytokine types. Investigation shows increased RORγt^+^ ILC3 cells enhanced lymph node metastasis [43,44]. However, as an immunological escape mechanism, TGF-releasing from the cancer cells changes NK cells to ILC1 cells in the TME [9]. Targeting NK cells in the TME may develop a new strategy for cancer treatment.

There has been a growing focus on developing small molecules targeting NK cells in the TME to enhance their activity against tumors. Studies have indicated that specific small molecule inhibitors like Ara-C, cisplatin, and fluorouracil (5-FU) can elevate the expression of NK cell activating ligands, resulting in improved recognition and elimination of cancer cells [45]. Likewise, bortezomib, a proteasome inhibitor used effectively in multiple myeloma treatment, has been found to induce the expression of ligands for NK cell activating receptors and augmenting NK cell response [46]. An immunomodulatory (IMiD) drug, lenalidomide, approved for multiple myeloma and myelodysplastic syndromes (MDS), impacts the immune response by increasing peripheral NK cell numbers. Lenalidomide potentially enhances NK cell activation by boosting ligand expression on tumor cells, prompting the release of NK cell stimulatory cytokines like T cell-derived IL-2 or even directly reducing the threshold for NK cell activation [47]. Although the exact mechanism behind lenalidomide’s impact on NK cells remains uncertain, recent combined studies with lenalidomide and rituximab have shown enhanced effectiveness against B cell malignancies by improving the antibody-dependent cell-mediated cytotoxicity (ADCC) effect [48,49]. Notably, combining lenalidomide with dexamethasone diminishes its immune-stimulating effect on NK cells, likely due to the suppression of IL-2 production in CD4^+^ T cells. Similarly, the small molecule ibrutinib interferes with rituximab’s ADCC effect in CD20^+^ B-cell lymphoma by irreversibly binding to IL-2 inducible tyrosine kinase (ITK), a crucial factor for FcR-stimulated NK cell function [50]. A comprehensive screening of natural products resulted in the identification of 20-deoxyingenol 3-angelate (DI3A) and its derivative ingenol 3-angelate (I3A) as a potent immune enhancer, which augments NK cell-mediated elimination of non-small cell lung cancer cells (NSCLCs). Further, validation demonstrated that DI3A and I3A augment degranulation and interferon-gamma secretion in NK cells [51]. Table 2 provides a comprehensive summary of diverse small molecules that specifically target various components within TME.

#### 3.2.2. Invariant Natural Killer T Cells

An evolutionarily conserved subgroup of NKT cells is invariant NKT (iNKT) cells. iNKT cells express semi-invariant T cell receptors (TCRs) that have invariant TCR α-chain rearrangements like V14-J18 (mice) and V24-J18 (human) along with a constrained number of TCR α-chains [89]. Both human and mouse iNKT cells are activated by α-GalCer and represented on the CD1d molecule like the MHC class I [90]. iNKT cells are noncirculating, and tissue-resident lymphocytes significantly vary between tissues. The subset makeup and tissue placement are crucial for iNKT cells to protect against dangerous infections and help maintain tissue homeostasis. iNKT cells are desirable immunotherapy targets due to the non-polymorphic restriction factor CD1d and the invariant characteristic of the iNKT cell TCR. Studies have shown that α-GalCer successfully modifies immune responses in various tumor models [91]. Several clinical trials demonstrated the benefit of activating iNKT cells to increase anti-tumor immunological response in cancer patients [92]. CD1d-expressing tumor cells are instantly recognized and eliminated by iNKT cells, particularly in prostate tumors, medulloblastomas, early myeloma, lymphomas, and myeloid leukemias [93].

#### 3.2.3. Natural Killer Cells

In 1975, the classification defined natural killer (NK) cells as distinctly divergent from T and B cells [94]. NK cells are crucial in the immunosurveillance of cancerous cells of their innate capacity to kill cancer cells without sensitization [95]. Additionally, NK cells are critical for preserving immunological homeostasis. During the immune response, NK cells receive signals directly or indirectly from other immune cells and cancer cells. Dendritic cells (DCs) type I IFN control the function and proliferation of NK cells by the secretory exosomes, trans-present IL-15, and IL-12 [96]. NK cells express NKG2D-like activated T lymphocytes and macrophages that the stress-induced ligands recognize and form a homodimer. NKG2D ligands include the expression MULT1 and REA-1 in mice and MICB, ULBPs, and MICA in human tumor cells. A metalloproteinase-dependent release of the NKG2D ligand from the cell surface is one method by which tumor cells generate an immune-invasive defense. NKG2D binding induces the activation of NK cells for cytokine generation and cytotoxicity activity [97]. Activated NK cells, T cells, and other lymphocytes express the co-stimulatory factor CD137 that promotes the growth of NK cells and IFN-γ production. Nearly all NK cell subtypes typically express TIM-3. Metastatic melanoma, gastric cancer, and lung adenocarcinoma patients demonstrated upregulated TIM-3 levels in NK cells. Galectin-9 (HMGB1) and ceacam-1 can pair with TIM-3 to modulate the NK cell function [98]. The upregulation of TIM-3 is associated with exhaustion of T-cells. However, its role in NK cells is still debatable [99].

NK cells also express several checkpoint proteins, including CD96 and TIGIT, representing ITIM motifs in their cytoplasmic domains. These proteins can identify CD112 and CD155 interchangeably with the NK cell activating receptor CD226. TIGIT and CD96 exhibit a greater propensity for CD155 than CD226 [100]. In vitro studies have shown that TIGIT blocking enhances NK cells’ cytolysis capacity. Furthermore, animals lacking CD96 had increased IFN-γ production of NK cells and the capacity to control tumor growth [101]. The cytokine-inducible SH2 (CIS) containing protein inhibits IL-15 signaling in NK cells. Truncation of the CIS gene increases survival, tumor cell cytotoxicity, and proliferation. Additionally, lowers the NK cell threshold to IL15. Therefore, CIS in NK cells could act as an intracellular checkpoint [102].

#### 3.2.4. Neutrophils

Neutrophils, the most prevalent immune cells, also have complex and significant roles in cancer [103]. Numerous studies have shown that individuals with various malignancies had higher peripheral blood neutrophil counts. The neutrophil-to-lymphocyte ratio (NLR) has been a reliable prognostic tool for cancer patients [104]. Different mice variants of KRAS-driven ovarian cancer show increased neutrophil-related chemokine levels and proliferation of neutrophils in the tumors and are associated with an immediate increase in neutrophil-related cytokines, including CXCL8 and GM-CSF [105]. The cytokine IL-8 and the chemokine ligands CXCL1, CXCL2, and CXCL5 draw neutrophils that overexpress CXCR2 to areas predisposed to malignancy. Bone marrow stromal cells regulate neutrophil retention and movement by producing and modulating the level of CXCL12, a ligand for CXCR4 and CXCR7. The expression CXCR4 inhibits CXCR2-mediated neutrophil trafficking from the bone marrow into the peripheral circulation. Neutrophil CXCR2 enhances neutrophil migration into areas that support cancer, while CXCR2 reduction in neutrophils enhances ROS generation and the pro-cancer effect [106]. The depletion or blocking of CXCR2 signaling in the neutrophil inhibits cancer development and suppresses angiogenesis [107]. Furthermore, utilizing an orally active small molecule antagonist SCH479833 to target the CXCR2/1 axis effectively suppresses neutrophil recruitment, angiogenesis, fibrosis, necrosis, metastasis, and overall tumor growth in preclinical murine models of pancreatic cancer [108]. These findings provide compelling evidence that the specific inhibition of CXCR2/1 via small molecule inhibitors holds significant promise as a therapeutic strategy for impeding cancer progression, including its growth, angiogenesis, and metastatic potential.

#### 3.2.5. Tumor-Associated Neutrophils

The TME’s key component, tumor-associated neutrophils (TANs), is actively implicated in the growth and spread of tumors. Numerous preclinical investigations have shown that TANs play pro-tumor activities by promoting extracellular matrix in the tumor environment and TME inflammation. TANs release growth factors like HGF that promote tumor growth and immunosuppressive substances like arginase-1 and TGF that inhibit adaptive immunity [109]. Circulating tumor cells and neutrophils interact in the bloodstream to promote tumor cell cycle progression, encourage extravasation of tumor cells, and dramatically increase CTC’s ability to metastasize [110]. High circulating NLR in the patients was associated with a poor prognosis of numerous cancer types, including liver, breast, and colon [111]. Neutrophil targeting has become a feasible treatment strategy for cancer. CXCR2 is the crucial regulator of neutrophil mobilization [112]. Targeting CXCR2 by small molecule inhibitors will provide a new therapeutic strategy for advanced cancer treatment.

#### 3.2.6. Tumor-Associated Macrophages

Tumor-associated macrophages (TAMs) engage in the development of the tumor microenvironment. Macrophages represent phenotypic heterogeneity and functional diversity essential in innate and adaptive immunity. It also upregulates tumor progression, invasion, metastasis, and drug resistance [113]. Both bone marrow and yolk-sac-generated monocytes engaged locally and differentiated into TAMs induced by growth factors and chemokines generated by stromal and tumor cells in the TME [114]. These macrophages are called M1-polarized macrophages and are activated by pro-inflammatory (IFN-γ) and immunostimulatory cytokines (e.g., interleukin (IL)-12 and IL-23)- express high levels of anti-tumorigenic molecules such as TNF-α, iNOS or MHC class II molecules. In contrast, M2 macrophages are classified as pro-tumorigenic and express high levels of arginase-1, IL-10, CD163, CD204, or CD206^.^ TAMs more closely resemble M2-polarized macrophages called alternatively activated macrophages. Th2 cytokines activate these (e.g., IL-4, IL-10, and IL-13). In addition, TAMs promote tumor proliferation, invasion, and metastasis. It also stimulates tumor angiogenesis and inhibits T cell-mediated antitumor immune response that promotes tumor progression.

Tumor-associated macrophages (TAMs) are prevalent in most cancers with poor clinical outcomes [113,115]. TAMs express a high level of immune checkpoint ligands, VISTA, PD-L1, and PD-L2, suppressing T cell activity and effector functions. The suppressed T-cell immunity by TAMs promotes Treg cell expansion in the TME [116]. Highly invasive cancer cells release high quantities of CCL2 and colony-stimulating factor 1 (CSF1) to attract and localize TAMs in metastatic locations. Further, it interacts with cancer cells to promote their intravasation, expansion, and migration [117]. TAMs are the presiding immunosuppressive cell types in the TME and are critical markers for the immunotherapy response. TAMs play an essential role in the therapeutic outcome of immunotherapy. It inhibits the anti-PD1 immunotherapy activity by preventing cytotoxic CD8+ T cell infiltration into the tumors [118]. TAM population gets elevated in tumors treated with anti-VEGF therapy, radiation, and chemotherapy.

Similarly, TAMs induce pro-inflammatory cytokines (IL-23, IL-6, and IL-1β), facilitating tumor development and progression. A recent study demonstrated that TAMs develop drug resistance to tumors by generating pyrimidine metabolites [119]. In contrast, inhibiting polarization or recruitment of TAMs increases the therapeutic efficacy, which is evidence of the defensive function of TAMs responding to immunotherapies [120]. Utilizing specific small molecule inhibitors to target TAMs could potentially establish an innovative approach for treating cancer.

The clinical trial involving the small molecule inhibitor RRX-001 on small cell lung cancer (SCLC) exhibited a reduction in SIRPα expression on macrophages and CD47 expression, resulting in potent antitumor activity and hypotoxicity [121]. PI3K-γ inhibitors like IPI-549 address immune checkpoint resistance by reshaping the tumor microenvironment, specifically by shifting macrophage polarization from M2 to M1 phenotype [122]. Treatment with small molecule R848 has been found to reprogram TAMs into the M1 type and enhance the ADCP effect, subsequently boosting the therapeutic anti-tumor impacts of TAMs [123]. For breast cancer models, blocking macrophage recruitment through CSF1R inhibition has proven to enhance gemcitabine’s efficacy in chemo-resistant transgenic pancreatic cancer models [124]. Furthermore, small molecule inhibitors that target CXCR2 on neutrophils and CCR2 on macrophages have demonstrated the ability to enhance the chemotherapeutic effects in models of pancreatic ductal adenocarcinoma. PLX-3397, a small molecule inhibitor of CSF1R, cKIT, and FLT3, has shown promise in reducing M2 macrophages and tumor burden when combined with adoptive cell transfer immunotherapy or other small molecule inhibitors. In clinical applications, the combination of the oral CCR2 small molecule antagonist PF-04136309 with conventional chemotherapy resulted in partial tumor responses (49%) and local tumor control in 97% of patients with advanced pancreatic ductal adenocarcinoma (PDAC) [125].

#### 3.2.7. Myeloid-Derived Suppressor Cells

A significant subpopulation of inhibitory immune cells is myeloid-derived suppressor cells (MDSCs) detected in tumors. They exhibit a changeable nature that makes their identification challenging [126,127]. Two primary subsets of MDSCs are monocytic MDSCs (mMDSCs) and polymorphonuclear or granulocytic MDSCs (pmnMDSCs). Phenotypically and morphologically, the mMDSCs are like monocytes, and the pmnMDSCs are like neutrophils. MDSCs have been impeded in cancer patients by cellular variety and the absence of markers. Modern definitions of human pmnMDSCs include CD11b^+^CD14^+^CD15^+^ or CD11b^+^CD14^+^CD66b^+^, while mMDSCs include CD11b^+^CD14^+^HLA^−^DR^−^/lowCD15^−^ markers [126,128]. The differentiation of MDSCs is favored by pancreatic CAFs, which release various soluble mediators, including IL-6, that promote the recruitment of MDSCs. CAFs utilize CXCL12 to attract monocytes to the TME and differentiate into MDSCs by IL-6-mediated STAT3 activation [129]. The MDSCs increasing impact on CAFs in breast cancer includes histone deacetylase-mediated epigenetic control [130]. Although ER stress shortens the lifespan of MDSCs, it may increase myelopoiesis and the MDSCs turnover in the patient. ER stress induces the apoptosis of MDSCs via activation of TRAIL-R or eIF2-ATF4-CHOP pathway. Exosomes produced by glioma stem cells inhibit systemic T-cells by directing CD14^+^ monocytes toward the mMDSC phenotype and upregulating IL-10 levels [131]. Similarly, MDSCs in the tumor-bearing hosts release immunosuppressive cytokine TGF-β [132]. Previous studies have identified that semaphorin 4D, tmTNF, and ribosomal protein S19 control MDSCs’ ability to produce TGF-β [133,134]. Additionally, PD-L1 regulates the MDSC-mediated T-cell suppression. PD-L1 blockade diminished the ability of MDSCs to inhibit T cells. Several studies reported that tumor-infiltrating MDSCs express more PD-L1 than their peripheral correlatives [135,136]. On the other hand, MDSCs display CTLA-4 activity. The precise regulation and function of CTLA-4 in MDSCs have not been fully explored. Reports suggested that the inhibition of CTLA-4 reduces MDSC frequency and ARG1 activity [137]. In addition, CSF-1 induces MDSC tumor invasion and combines the treatment of anti-CTLA-4 with CSF-1/CSF-1R to inhibit MDSC signaling [138].

Utilizing small molecule inhibitors to target MDSCs has shown remarkable potential in combating tumor growth. For instance, sunitinib, a small molecule inhibitor, has exhibited inhibitory effects on MDSC accumulation and suppressive activity in RCC tumor-bearing mice and other malignancies like colon and breast cancer. This inhibitor has specifically demonstrated the reduction in monocytic MDSC proliferation and induction of apoptosis in PMN-MDSCs [139]. Additionally, tyrosine kinase inhibitors (TKIs) such as dasatinib, nilotinib, and sorafenib have been explored to hinder MDSC differentiation, reducing MDSC-mediated suppression of CD8^+^ T cells. While sorafenib, a multi-kinase inhibitor, is recognized as a primary therapy for advanced liver cancer, it has also effectively diminished Tregs and CD11b^+^GR-1^+^ MDSCs in tumor-bearing mice [140]. Axitinib, a selective TKI, has shown promising outcomes by decreasing suppressive monocytic (MO)-MDSC activity and increasing immune cell populations in melanoma models. SAR131675, a VEGFR-3-specific TKI, has displayed potential in promoting M1-type TAMs and restraining MDSCs in the context of tumor progression [141].

Similarly, the FGFR inhibitor ZD4547 has proven effective in reducing tumor-associated MDSC populations [142]. JAK-1 inhibitor GLPG0634 and JAK3 inhibitor PF-956980 have been observed to induce MDSC differentiation, and AZD1480 administration has shown reduced MDSC populations and enhanced suppressive activity on T cell proliferation [143]. Flavopiridol, a pan-CDK inhibitor, has shown contrasting effects, elevating the levels of CX3CR1 expression and exacerbating tumor growth due to MO-MDSC accumulation [144]. A tolerable selective STAT-3 inhibitor, S3I-201, has successfully diminished MDSC populations in mouse models with head and neck squamous cell carcinoma [145]. Moreover, specific inhibitors like J32, targeting PI3Kα, exhibit cytotoxicity against PMN-MDSCs, and the combination treatment of J32 with immune checkpoint blockade antibodies results in significantly reduced PMN-MDSCs and tumor growth. Combining IPI-145 with the immune checkpoint inhibitor PD-L1 mAb displays a decrease in PMN-MDSC suppressive activity, subsequently improving CD8^+^ T cell activation and survival in oral cancer models [146]. These findings underscore the potential of small molecule inhibitors in modulating MDSC functions to enhance immune responses against tumor growth.

#### 3.2.8. Dendritic Cells

The Dendritic cells (DCs) are members of myeloid cells with diverse morphologies, ontogenies, and immunological characteristics [147]. DCs are critical for promoting and regulating the immune response to cancer. In tumors, DCs infiltrate into the TME and encounter tumor antigens expressed by tumor cells. DCs are vital antigen-presenting cells efficient in capturing, processing, and presenting the tumor antigens to T cells [148]. Mature and activated DCs neutralize the tumor antigens by presenting tumor antigens to T cells. This presentation promotes the activation of T cells and the elimination of tumor cells. However, tumors manipulate the function of DCs in the TME and develop an immunosuppressive environment to promote tumor progression. The factors released by tumors, such as cytokines (TGF-β, IL-6, and IL-10) and chemokines (CCL2 and CCL22), inhibit DC activation or recruitment. These factors promote the differentiation of DCs into regulatory HLA-G^+^ DC10 that suppress T cells’ immune responses [149]. Tumors engage immunosuppressive cells such as MDSCs or Treg cells to inhibit the activation and function of DCs. Treg cells inhibit the maturation and antigen presentation of DCs. Similarly, MDSCs suppress DC activation by producing immunosuppressive molecules such as arginase and reactive oxygen species [150,151].

Strategies to enhance DC function in the TME are a progressive area of research in cancer immunotherapy. These include DC-based vaccines, which involve loading DCs with tumor antigens and injecting them into the patient to activate an immune response against the tumor. DC-based vaccines have shown promising outcomes in clinical trials to activate an immune response against tumors. DCs can be used as a biomarker to predict patient prognosis and response to treatment. The research focuses on developing new strategies for efficient DC maturation and recruitment in the TME. The functionally active DCs may more effectively present tumor antigens to T lymphocytes and activate an immune response against the tumors. Combining DC-based therapies with chemotherapy or radiation therapy may enhance their efficacy and efficiently target tumor cells. In addition, several small molecules have been identified as efficient therapeutic agents to boost the function of DCs in tumors (Table 2).

#### 3.2.9. B Cells

Bone marrow-derived B cells are crucial for immunity, producing antibodies against tumor antigens. Activated B cells stimulate NK cells and macrophages for tumor elimination and present antigens to T cells, sparking antitumor responses. Within TME, B cell subpopulations naive, memory, and regulatory B cells (Bregs) have diverse effects. Depending on location and activation, B cells can promote or hinder tumor growth. Their full role in TME remains unexplored, but studies underline their importance in immunity and how B cell heterogeneity impacts cancer response. Bregs, a novel B cell subset, secrete IL-10 and TGF-β to dampen immune responses and boost tumor growth [152]. They influence TME by interacting with immune cells, regulating their function, and displaying immunological checkpoints (PD-L1) to inhibit NK and effector T cells. Bregs drive metastasis by converting CD4^+^ T cells into Tregs. CD5^+^ B cells in tumors engage IL-6, activating STAT-3 and JAK-2 pathways [153].

BTK (Bruton’s tyrosine kinase) is a crucial cytoplasmic kinase in B cell antigen receptor signaling, regulating pathways like PI3K/AKT/mTOR, MAPK, and NFκB [7]. As these pathways govern B-cell processes, BTK inhibition is crucial in treating B-cell tumors and immune diseases with minimal toxicity. BTK inhibitors are small molecules targeting BTK’s activity. They include the first approved Ibrutinib, which covalently binds BTK to halt its kinase activity in malignancies like CLL and MCL. Similarly, acalabrutinib selectively targets BTK, approved for CLL and B cell lymphomas, with favorable safety and efficacy profiles. Zanubrutinib is another BTK inhibitor approved for certain lymphomas. Like acalabrutinib, it is designed for higher selectivity toward BTK, potentially reducing off-target effects. Tirabrutinib and orelabrutinib, classified as second-generation BTK inhibitors, exhibit significant impact. Tirabrutinib’s binding to BTK with an IC_50_ of 2.2nM shows favorable efficacy in phase II trials for WM and primary CNS lymphoma patients. Orelabrutinib has gained approval in China for MCL and CLL/SLL patients with prior treatment [154,155]. These inhibitors target BTK at Cys481, with mutations causing resistance due to disrupted covalent binding. Third-generation inhibitors, namely pirtobrutinib and nemtabrutinib, bind reversibly and non-covalently to BTK kinase by passing resistance from prior inhibitors. Pirtobrutinib effectively overcame C481S and C481R resistance, as demonstrated in phase I/II trials (NCT03740529) for various B-cell malignancies. The ongoing phase III trials (NCT05023980, NCT04965493, NCT04662255) indicate promising potential for pirtobrutinib against multiple B-cell malignancies [7]. Similarly, nemtabrutinib is in early-stage clinical trials (NCT04728893, NCT05458297) for B-cell malignancies. While not a small molecule inhibitor, liso-cel represents a CAR T cell therapy targeting CD19 on B cells. When combined with BTK inhibitors, such as those described above, liso-cel can augment its therapeutic efficacy, illustrating the potential synergistic impact of these approaches in cancer treatment.

Targeting B-cell-related cytokines and pathways with small molecules can enhance B-cell recruitment to the TME and bolster antitumor responses. Cytokine-driven signaling, such as the CXCL13/CXCR5 and CCL19, 21/CCR7 axis, facilitates B cell and tertiary lymphoid structure (TLS) accumulation in tumors, impacting survival outcomes. Notably, CXCL13 and CCL19, 21/CCR7 can enhance immune infiltration in TME [156]. Nevertheless, the dual nature of these cytokines, demonstrated in various tumor types, underscores the importance of nuanced targeting. Strategies like lentiviral vectors encoding co-stimulatory ligands, such as CD40L, CD70, OX40L, or 4-1BBL, have shown potential in promoting potent B cell co-stimulation and cytotoxic activities, enhancing survival rates. These innovative approaches hold promise for amplifying B-cell-based immunity and mitigating their hidden protumor effects, providing novel avenues for therapeutic intervention in cancer treatment.

#### 3.2.10. Helper CD4^+^ T Cell

Helper T cells consist of two subtypes, T helper type 1 (TH1) and T helper type 2 (TH2), which interact molecularly with several immunological signaling pathways. The immunological actions and cytokine profiles of the CD4^+^ TH cells polarize into one of the following effector types: Tfh, Treg, TH17, TH9, TH2, and TH1. TH1 and TH2 cells are distinguished by the production of IL-4 and IFNγ, respectively. In addition, Tbet promotes TH1 differentiation by inhibiting TH2/TH17 differentiation and is expressed when IL-12 attracts NK cells to create IFN-γ. Together, these events activate the STAT-1 and STAT-4 signaling pathways. It has been shown that the cytokines released by TH2 cells, including IL-17, IL-13, IL-10, IL-5, and IL-4, inhibit tumor growth [157]. The small molecule BX-795 was observed to elevate IL-2 levels in culture supernatants of various T cell types. This effect occurred following TCR-dependent and allergen-specific stimulation. Remarkably, BX-795 also demonstrated the capability to suppress the secretion of Th2 cytokines simultaneously. Furthermore, BX-795 exhibited significant inhibitory effects on Th2-related inflammation, including the expression of transcription factors and cytokines associated with Th2 responses, and it reduced the infiltration of type 2-associated inflammatory cells like eosinophils into the lungs [158].

CD4^+^ T cells can differentiate into diverse functional subtypes, facilitating their collaboration with appropriate effector immune cells for immune responses. Early research suggested that CD4^+^ T cell-induced cytokine release, notably IL-2, transmitted signals to nearby CD8^+^ T cells interacting with the same dendritic cell. Conventional type 1 dendritic cells (cDC1) respond to activated CD4^+^ T cells by generating cytotoxic T lymphocyte (CTL) responses against cell-associated tumor antigens [159]. The newly identified small molecule 6809-0223 enhances IL-2 secretion by CD4^+^ and CD8^+^ T cells, promoting increased CD4^+^ T cell proliferation in murine models. These findings prompt the exploration of small molecules to augment IL-2 production in CD4^+^ T cells, potentially enhancing robust anticancer immune responses [160]. Furthermore, there have been efforts to engineer modified IL-2 molecules that selectively enhance CD4^+^ T cell responses while minimizing undesirable effects on regulatory T cells (Tregs), which can suppress immune responses [161]. A recent study conducted by Borodovsky et al. offers evidence supporting adenosine’s role in immune evasion by tumors, dampening both adaptive and innate immunity. They found that targeted small molecule inhibition of A2AR using AZD4635 counteracts adenosine signaling, restoring T cell activity via an internal mechanism. Additionally, this inhibition enhances tumor antigen cross-presentation by CD103^+^ DCs, reinvigoration antitumor immune responses [162].

Chronic exposure to tumor antigens induces dysfunction in CD4^+^ T cells, distinct from the exhausted phenotype observed in CD8^+^ T cells. Both exhausted CD4^+^ and CD8^+^ T cells express similar coinhibitory receptors. In the late activation stages, dysfunctional CD4^+^ T cells produce more CTLA-4 than their CD8^+^ counterparts. Via CD40/CD40L interactions with dendritic cells, these cells serve as “helpers” for cytotoxic CD8^+^ T cells [163]. Tumor antigen-specific CD4^+^ T effector cells play a crucial role in maintaining anti-tumoral immune responses, producing important cytolytic granule molecules like Gzm and perforin; CD8^+^ T cells direct toxicity, contributing to protective and pathogenic immunity. To restore CD4^+^ T cell functionality, small molecules targeting co-stimulatory receptors like CD40, 4-1BB, and OX40 have been explored. Notably, small-molecule OX40 modulators (DB36, CVN) exhibit partial agonist properties, demonstrating the feasibility of modulating OX40-OX40L interaction [164]. These molecules also have the potential to enhance interactions between CD4^+^ T cells and antigen-presenting cells, thereby promoting improved immune responses [165].

Efforts to invigorate CD4^+^ T cell responses against tumors have focused on peptide epitope immunization, aiming to generate TH1-polarized CD4^+^ T cells [166]. CD4^+^ CTLs possess both cytokine release and cytotoxic capabilities for individuals with MHC class I presentation deficits, potentially compensating for CD8-mediated neutralization. Studies highlight that engineered human CD4 CD26 ^high^ T cells, armed with a mesothelin-chimeric antigen receptor, elicit heightened immune responses against successful mesothelioma in NSG mice post-adoptive transfer, surpassing other helper CD4 subsets with the same CAR. Notably, tumor regression resulted from adoptive cell therapy (ACT) using endogenous tumor-infiltrating CD4^+^ T lymphocytes recognizing a mutant ERBB2 protein in a patient with metastatic epithelial carcinoma. TCR-engineered CD4^+^ T cells carrying an HLA-DP4-restricted TCR for the cancer-testis antigen MAGE-A3 demonstrated safety and efficacy in inducing objective tumor regression across multiple cancer types, validating the potential of modified MAGE-A3-specific CD4^+^ T cells [167].

#### 3.2.11. Cytotoxic CD8^+^ T Cell

Cytotoxic CD8^+^ T cells (CTLs) provide adaptive T cell immunity along with the sister lineage of CD4^+^ cytotoxic T cells. CTLs are incredibly adaptable and polyfunctional cells. CD8^+^ T cells activated and primed toward effector CTLs to provide long-lasting and effective antitumor immune responses. The primary goal of CD8^+^ T cells is to prime and strengthen the relationship between CD4^+^ T cells in innate and adaptive immunity cells such as NK and DC cells. Checkpoint receptors like PD-1/PD-L1 and CTLA-4 can be targeted to prevent CD8^+^ T lymphocytes from being exhausted and restore their priming, which helps to destroy the cancer cells producing antigens. Alternatively, the small molecule NSC622608 demonstrated effective inhibition of VISTA signaling, leading to increased T-cell proliferation and the re-establishment of T-cell activation. Notably, this effect was observed even when cancer cell lines expressing VISTA were present. These findings underscore the potential of small-molecule compounds targeting VISTA as promising immunomodulatory agents [168].

EGFR inhibitors enhanced basal and IFN-induced presentation of MHC class I, improving CD8^+^ CTL function and destroying tumor cell targets [169]. Cancer stem cell (CSC)-like subpopulations also show mesenchymal characteristics and differentiate into other lineages resembling CD8^+^ CTL responses [170]. Since B7-H3-expressing NSCLCs prevents CD8^+^ T cell-mediated immune surveillance, developing a new therapeutic modalities combination of anti-PD-1/PD-L1 with anti-B7-H3 immunotherapy to activate the anti-tumor activity of CD8^+^ T cell [171]. A recent study uncovered the potential of ipilimumab and tremelimumab to augment the infiltration of intratumorally CD4^+^ and CD8^+^ cells in human tumors while preserving the FOXP3^+^ cell population. This finding implies that co-administering these treatments with a B7x inhibitor might offer a viable strategy to counteract B7x-mediated resistance to anti-CTLA-4 therapy, opening new avenues for enhanced therapeutic outcomes [172].

Moreover, there was a notable increase in the expression of the H3K9me3-specific histone methyltransferase SUV39H1 in human colon carcinoma. This enzyme plays a role in suppressing the expression of CD8 CTL effector genes, contributing to immune evasion by colon tumors. An innovative study by Lu et al. in 2019 introduced a novel small molecule, F5446, designed to target H3K9me3 and reduce SUV39H1 expression on T cells, thus enhancing their anti-tumor activity [173]. This highlights the potential of F5446 to modulate effector gene expression and bolster the function of tumor-infiltrating CTLs. Additionally, recent research introduces the antitumor immune modulator tubeimoside-1 (TBM-1) as a negative regulator of PD-L1 levels. TBM-1 fosters PD-L1 lysosomal degradation in a TFEB-dependent manner, independent of autophagy. By binding selectively to the mammalian target of rapamycin (mTOR) kinase, TBM-1 inhibits mTORC1 activation, leading to TFEB nuclear translocation and lysosome biogenesis [174].

This study highlights the synergy between TBM-1 and anti-CTLA-4 treatment, effectively amplifying antitumor T cell immunity while mitigating immunosuppressive MDSC and Treg cell infiltration. This uncovers a previously unknown antitumor mechanism of TBM-1, offering an innovative approach to augment the effectiveness of cancer immunotherapy through immune checkpoint blockade. Conversely, applying 4-1BB aptamer-raptor siRNA demonstrated reduced mTORC1 activity in numerous CD8^+^ T cells. This led to robust memory responses, enhanced cytotoxic functions, and improved vaccine-induced protective immunity against tumors [175].

Therapeutic candidates targeting the epigenetic regulatory family of histone deacetylases (HDAC) have shown clinical benefits in treating some hematologic malignancies. For example, class I HDAC inhibitors upregulated the expression of PD-L1 and PD-L2 in melanomas and enhanced PD-1 blockade immunotherapy [176]. Mechanistically, the treatment of HDAC inhibitor demonstrated rapid elevation of histone acetylation of the PD-L1 gene to promote long-term PD-L1 expression. These results highlight the ability of epigenetic modifiers to elevate the activity of immunotherapies, providing a possibility for a combination of small molecules and HDAC inhibitors with PD-1 blockade to augment the antitumor activity of CD8^+^ CTL [177].

In progenitor-exhausted CD8^+^ T cells, lysine-specific histone demethylase 1 (LSD1) works to impose an epigenetic program to counteract TCF1’s progenitor maintenance and induce terminal differentiation. The persistence of progenitor-exhausted CD8^+^ T cells is increased by genetic modification or small molecules targeting LSD1. These cells serve as a consistent source for the proliferative transformation to quantitatively more comprehensive terminally exhausted T cells with tumor-killing cytotoxic effects, which results in effective and long-lasting responses to anti-PD1 therapy [178]. On the other hand, a combination of Dacarbazine and PSB1115, an A2BR antagonist, exhibited enhanced infiltration by CD8^+^ T cells and NKT cells. Moreover, these tumors displayed elevated levels of GzmB compared to the tumors that received Dacarbazine alone. Similarly, the simultaneous administration of AB928, acting as both an A2AR and A2BR antagonist, alongside Doxorubicin or Oxaliplatin, amplified the intra-tumoral presence of tumor-specific CD8^+^ T cells [179].

Recent research has highlighted the potential for reversing CD8^+^ T cell exhaustion to bolster anti-tumor responses. A groundbreaking study identified 19 compounds from the ReFRAME drug-repurposing collection that effectively restore cytokine production and enhance the proliferation of exhausted T cells. Notably, ingenol mebutate, a protein kinase C (PKC)-inducing diterpene ester, emerged as a key player in counteracting the suppressive signaling cascade mediated by IR signaling on T cells [180]. These findings provide a disease-relevant strategy for identifying T cell function modulators and uncovering new targets for immunotherapy. Similarly, Hematopoietic progenitor kinase 1 (HPK1 or MAP4K1) has been recognized as a negative intracellular immune checkpoint in antitumor immunity studies involving HPK1 knockout and kinase-dead mice. To explore HPK1’s role in human immune cells and its therapeutic implications, researchers sought pharmacological HPK1 inhibition. A recent study unveiled a novel, potent, and selective HPK1 small molecule kinase inhibitor, termed compound K (CompK) [181]. CompK treatment notably enhanced human T-cell functions, improving TCR avidity for tumor-associated antigens (TAAs) and boosting the cytolytic activity of CD8^+^ T cells against tumors. Moreover, CompK facilitated dendritic cell maturation and priming in tumor-draining lymph nodes, making it a promising pharmacological tool to address cancer treatment resistance.

#### 3.2.12. Tissue-Resident Memory T Cells

Tissue-resident memory T cells (TRM) are protective against subsequent antigen exposure or vaccination. Self-renewing, long-lived TRM cells remain stationed within tissues without recirculating. Found in various tissues, including skin, gut, and lungs, they prevent tissue egress via CD103, CD49α, and CD69 markers. Specific transcription factors like Runx3, Notch, and BATF define their profile. Specialized for local immune surveillance, TRMs respond rapidly to re-exposure, impacting both infectious disease and cancer immunity, showing potential in immunotherapy responses [182,183].

TME comprises a dynamic structure with immune cells like CTLs, TAMs, and Treg cells. CD8^+^ CTLs are crucial anti-tumor effectors, making them prime candidates for immunotherapy. Tumor-infiltrating lymphocytes (TILs), specifically CD8^+^ T cells expressing CD103 and CD69, have gained attention. CD103 activates T cells within the TME, along with specialized chemokine receptors like CXCR3 and CCR5, producing chemokines such as CCL3, CCL4, and CCL5. CD103-expressing CD8^+^ T cells are linked to patient survival in bladder, lung, and ovarian cancer. NKG2A, an inhibitory receptor in the KIR family, is expressed in T and NK cells. Tumor-infiltrating NKG2A^+^ CD8^+^ T cells, prominent in lung cancer, display TRM and exhausted T cell markers, suggesting a target for efficient anti-lung cancer immunotherapy [184,185].

Utilizing small molecules to target TRM cells for enhanced function and fitness within the TME holds promise in cancer immunotherapy. TRM cells release vital effector molecules and cytokines like IL-2, IFN-γ, TNF-α, and GzmB, enabling rapid responses against cancer cells [186]. Targeting inhibitory and costimulatory receptors on CD8 TRM cells is a potent strategy. For instance, small molecules Ingenol and Mebutate override suppressive signaling via immune checkpoints, boosting activation and function [180]. Similarly, F5446, a histone methyltransferase inhibitor, may enhance TRM activity by reducing SUV39H1 expression [180]. Tubeimoside-1 negatively regulates PD-L1 to enhance TRM cell activity [186]. Small molecules degrading inhibitory receptors (PD-1, PD-L1, TIM-3) and promoting fatty acid metabolism (PPAR-γ agonists) may encourage and sustain TRM fitness [186]. Additionally, applying Quinazolinamine derivatives activates DC cells by enhancing antigen cross-presentation, which can boost TRM activity [187]. Discovering small molecules that enhance transcription factor activity (e.g., Bhlhe40) and increase soluble factors (TGF-β, IL-2, IL-15) presents a unique avenue for novel anti-cancer therapies [180,188]. These instances underscore the capacity of small molecules to enhance CD8^+^ TRM cell function within cancer immunotherapy. Continuous research holds the promise of unveiling additional molecules with similar potential.

#### 3.2.13. Regulatory T Cells

Immunosuppressive FoxP3^+^ CD4^+^ regulatory T cells (Tregs) are crucial immune components within the tumor microenvironment (TME), initially contributing to immune homeostasis and self-tolerance. However, TME Tregs hinder effective immune responses against cancers, curbing protective immunosurveillance and impeding anticancer immunity. Tregs can differentiate from conventional T cells within TME, gaining strong immunosuppressive activity that suppresses antitumor responses and fostering tumor progression [189]. Forkhead box P3 (FoxP3), a distinct Treg marker, dominantly controls Treg function, maintaining suppressive capacity during autoimmunity and infections. Studies indicate FoxP3-expressing Tregs’ prognostic significance across various cancers. A low CD8^+^ T cells to Tregs ratio correlates with poorer bladder, ovarian, and breast cancer outcomes [190]. Tregs, like CD8^+^ T cells, infiltrate tumor stroma, whereas CTLA4-expressing Tregs hinder T cell activation. Persistently high IL-2 receptor expression in Tregs disrupts IL-2 availability for CD8^+^ T cell development, further hampering antitumor immunity [191]. Moreover, in pancreatic cancer models, Tregs indirectly suppress CD8^+^ T cell activation by restraining tumor-associated dendritic cell growth and immunogenicity, curtailing IFN release and impairing tumor control. Denileukin diftitox, an FDA-approved small molecule used to treat CD25^+^ cutaneous T-cell leukemia and lymphoma, was assessed for its potential in lung, ovarian, and breast cancer treatment. The study demonstrated that a sole administration of this molecule resulted in reduced frequency and total count of peripheral Tregs in all patients, accompanied by enhanced activation of effector T-cells [172].

Moreover, it has been suggested that Treg cells decrease immune response by several different methods, including directly limiting effector T cells, suppressing co-stimulatory signals by CD86 and CD80 via CTLA-4, and inhibitory cytokines. However, despite Tregs being often protumoral, some research has shown that the prevalence of tumor-infiltrating Tregs indicates a favorable prognosis in Hodgkin’s lymphoma and colorectal cancer [192]. Due to the complex relationship between CTLs and Tregs, it may be possible to reduce Treg activity to boost CTL production and restore their ability to make cancer cells more immunosensitive [193]. In CARM1-deficient tumors, there were significantly more NK, CD8^+^ T cells, and DC-invading CD8^+^ T cells expressing few exhaustion markers. A small molecule that targets CARM1 produces powerful antitumor immunity and makes resistant tumors susceptible to checkpoint inhibition. Hence, targeting this co-transcriptional regulator can improve immune function while converting the resistant tumor cells to be more vulnerable to immune attack [177].

Several critical processes that affect T cell immunological responses are regulated by proteases, and some of them, including asparagine endopeptidase, furin, MALT, and ADAM, are involved in Treg immunobiology. Targeted protease inhibition has been shown to specifically modulate Treg activity in experimental mouse models, either with small molecule inhibitors or gene knockout mice [194]. JG-1 inhibits TGF-1 expression and the differentiation and invasion of Treg cells, promoting the differentiation of Th17 cells and increasing CCR6 and CCL20 expression [195]. The connection between TNF and TNF receptor type II (TNFR2) is crucial for suppressive Tregs’ phenotypic consistency, growth, and activation [196]. It is possible to utilize an immunotherapy drug that targets the Breg–Treg axis to inhibit T cells via the TGF-mediated pathway used by tumor-evoked Breg cells [197]. Engineered STAT-3-silenced T cells or small molecule inhibitors that target STAT-3 systemically can both be used to grow and activate adoptively transplanted T cells in vivo [198]. The immune system is regulated by controlling cell chemotaxis, and during acute or chronic systemic inflammation, the regulatory Treg subsets and pro-inflammatory Th17 are selectively delimited. It is suggested that inhibiting CCR6 can lessen disease symptoms and aid in the recovery of various inflammatory and autoimmune conditions [199].

Inhibition of haspin kinase activates cell-intrinsic and extrinsic anti-tumor activity. CX-6258’s target, haspin kinase (HASPIN), has been determined. A mouse model study shows that CX-6258, a HASPIN inhibitor, reduces proliferation, frequent production of micronuclei, and activation of the STING pathway. Additionally, it increases the number of IFN-producing CD8^+^ T cells and lowers the frequency of Treg cells in vivo. It also causes a robust cGAS-dependent type-I interferon response in tumor cells [200]. Small molecules that target HASPIN may, therefore, overcome melanoma treatment resistance, control the TME, and target vulnerabilities in several cancer lineages. Small molecule inhibitors of the tyrosine kinases c-met “aka cabozantinib, altered the frequency of immunological sub-populations in TME and at the periphery, creating a more hospitable immune milieu. Cabozantinib dramatically decreased the activity of regulatory T cells and enhanced the release of cytokines from effector T cells in response to the antigen when coupled with a cancer vaccine made from a poxvirus that targets a self-antigen. Direct manipulation of tumor cells and these changes to the immunological environment resulted in significantly increased antitumor efficacy [201]. Many cancer forms have been discovered to have high levels of CL2, which has been linked to a bad prognosis. Hence, CCL2/CCR2 signaling axis inhibition or blocking has been of interest for cancer therapy.

Anti-PD-1 therapy alone does not sensitize tumors, as well as anti-PD-1 therapy in combination with CCR2 antagonism. Increased CD8^+^ T cell activation and recruitment and a concurrent decline in CD4^+^ regulatory T cells are correlated with improved treatment response. These findings offer compelling preclinical support for further clinical investigation of PD-1/PD-L1-directed immunotherapies along with CCR2 antagonism against several tumor types, particularly in light of the accessibility of CCR2 small molecule inhibitors [202]. Similarly, utilizing a blend of Tregs depletion via anti-CTLA-4 and PD-1 blockade presents a potential approach to bolster CD8 activation and Treg reduction [172]. A2ARs are potent inhibitors of activated T cells’ antitumor and pro-inflammatory activities. The most significant therapeutic targets for this axis are CD39/CD73, A2ARs, and HIF-1α. The efficiency of contemporary immunotherapeutic strategies against cancer, such as CTLA-4/PD-1, is increased by downregulating this axis. A2AR is severely inhibited by AZD4635, which reduces cancer volume and boosts anticancer immunity. Both human and mouse CAR T cells become significantly more effective when the A2AR gene is deleted using CRISPR/Cas9 [203]. B7-H4 is crucial for immunological control because it prevents T cells from proliferating to produce a negative regulatory T-cell immune response. Both in vivo and in vitro, B7-H4 caused CD8^+^ T cell mortality, had an impact on the PD-1/STAT-3 pathway, and encouraged tumor cells to evade the immune system [204]. The key transcriptional regulator of CD8^+^ Tc17 and CD4^+^ Th17 cell development and function is RORγt. RORγ^+^ cells comprise 15% of the CD4^+^ T cell fraction in lymphocytes that have infiltrated various human tumor types. By inhibiting Treg development, lowering CD73 and CD39 expression, and lowering expression of coinhibitory receptors such as TIGIT and PD-1 on tumor-reactive cells, RORγ agonists also lessen immunosuppressive mechanisms [205].

## 4. Hybrid Cells or New Immune Cells “Soldier”

Cellular transformation activates phenotypically diverse subsets of conventional tumor-infiltrating T cells and nonconventional innate-like T cells (ILT) [206]. The fusion of different tumor cells or with normal cells develops newly discovered tumor hybrid cells (HCs) that display novel properties such as increased proliferation, migration, drug resistance, decreased apoptosis rate, and avoidance of immune surveillance [207]. Evidence demonstrated that HCs are associated with a high frequency of cancer metastasis. The potential mechanisms involved in the formation of HC are matrix metalloproteinase-9 (MMP-9), pro-inflammatory cytokine tumor necrosis factor-α (TNF-α), and syncytia A mucogenic proteins along with the membrane phospholipid, phosphatidylserine (PS) [208]. In addition, viruses trigger the cell fusion of infected and uninfected cells by producing viral fusogens for HC formation [209].

Several types of tumor hybrid cells arose from the fusion of cancer cells with stromal cells, fibroblasts, MSCs, and immune cells (Figure 3). The phenotype and the role of HCs in cancer have not been explored yet. Extensive investigations focusing on the comprehensive and multi-omics analysis of HCs will be required to identify the genetic alterations of tumor cell fusion that led to the formation of HCs. It also identifies the specific markers of HCs and their correlation with clinical and pathological parameters in various tumors. In addition, the genetic and proteomic information will provide strategies to block tumor cell fusion, therapy resistance, and tumor metastasis by targeting HCs for advanced cancer treatment. Further, targeting MMP-9 for the TNF-α-induced fusion of epithelial and triple-negative breast cancer cells, Toll-like receptors (TLRs), and Wnt/β-catenin signaling pathway will be critical to prevent cell fusion and tumor growth **(**Figure 3) [207]. For example, small molecule inhibitor XAV-939 targeting the Wnt/β-catenin signaling pathway suppresses macrophage’s fusion capacity with tumor cells, inhibiting cell proliferation, migration, and invasion [210]. Thus, targeting HCs with small molecules could be a potential therapeutic option for cancer treatment.

Cancer immune surveillance is contingent upon the function of cytotoxic T cells, which infiltrate tumor sites and identify and eliminate cells undergoing malignant transformation. However, the effectiveness of tumor-infiltrating classical αβ T cells can be compromised, leading to a state of dysfunction termed ‘exhaustion,’ driven by persistent exposure to mutation-generated neoantigen peptides distinguishing tumor cells from their regular counterparts [211]. Unlike classical T cells, innate T cells express T-cell receptors (TCRs) that do not typically recognize major histocompatibility complex (MHC)-peptide complexes. As a result, they demonstrate the capacity to sustain an activated state while evading exhaustion [212]. Pioneering investigations led by Chou et al. have ignited substantial curiosity regarding the prospective anti-tumor capabilities exhibited by a specific subset of T cells categorized as ‘innate-like T cells [213,214]. Thymic progenitors generate conventional CD4^+^ and CD8^+^ T cell subsets and the unconventional innate-like T cell (ILT) subsets (Figure 4).

Several types of ILT have been identified, including (i) CD8αα ILTCKs, (ii) CD8αα intraepithelial lymphocytes (IELs), (iii) γδ T cells, (iv) iNKT cells, (v) mucosal-associated invariant T (MAIT) cells, and (vi) CD4^−^CD8^−^ T cells. These ILT subsets recognize non-peptidic ligands, such as phosphorylated metabolites for γδ T cells, glycolipid antigens that interact with iNKT cells expressing an invariant αβ TCR in conjunction with the non-classical class I molecule CD1d and riboflavin metabolites binding to the αβ TCR of MAIT cells through the non-classical class I molecule MR1 [212,215]. Accumulating evidence indicates the infiltration of these diverse innate T cell subsets within tumors, where they can eliminate transformed cells and influence adaptive immune responses. This cumulative impact directly affects tumor development and growth. Both pro- and anti-tumorigenic functions have been attributed to NKT and γδ T cells, demonstrating their direct and indirect influence on tumor growth. Remarkably, MAIT cells have also been observed infiltrating tumors and contributing to tumor cell eradication [216]. Furthermore, unconventional CD4-CD8- TCR αβ T cells have been demonstrated to mediate an IFN-γ response and impart tumor resistance in a sarcoma model [217]. In summary, the emergence of these innate-like T cell subsets from thymic progenitors highlights their pivotal role in immune surveillance within the intricate tumor microenvironment. Enhancing the function of ILT cells through small molecules, diverse interactions, capacity to influence tumor progression, and potential to modulate adaptive immune responses positions them as captivating prospects for novel therapeutic strategies to combat cancer.

Adding to the diverse repertoire of innate-like T cell subsets, a recent breakthrough has uncovered a novel, evolutionarily conserved class of αβ T cell receptor (TCR)-positive cells. These newly identified cells, termed “Innate-like T cell killers” (ILTCKs), express the FCER1G gene and exhibit a remarkably high cytotoxic potential [213]. The seminal work by Chou and colleagues has shed light on the unique attributes of this population, revealing heightened cytotoxicity, resilience against exhaustion, and a particularly noteworthy enhancement in tumor-homing capacity when compared to conventional cytotoxic (aka “killer”) T cells [213]. These cytotoxic ILT cells were detected for the first time in murine prostate and human colorectal cancer tissues characterized by high NK1.1 expression [213,218].

Tbet, a master transcription factor, and IL-15 play crucial roles in ILTCK development and maintenance, distinct from Nfil3’s influence on NK cells [214]. Single-cell RNA-seq of PyMT tumor CD8^+^ T cells revealed unique signatures for early effectors (Il7r, Tcf7), exhausted (Pdcd1, Tox), and ILTCKs (GzmB, Klrb1c, Fcer1g). This ILTCK signature was consistent in prostate cancer models and colorectal cancer patients. TCR analysis indicated polyclonality of ILTCK TCRs, contrasting the oligoclonal nature of PD1^+^ T cells [213,214]. ILTCK ab TCRs exhibited heightened reactivity against shared antigens, suggesting broader antigen recognition. MHC class I restriction and peptide-MHC complex were crucial for ILTCK activation. Separate αβ TCRs implied distinct origins for CD8^+^PD1^+^ and ILTCKs. Transcriptional profiles revealed antigen-related genes in ILTCK progenitors, suggesting autoantigen-driven selection, with Fcer1g emerging as a lineage-defining marker for ILTCK, already upregulated in thymic progenitors and expressed in differentiated and activated tumor-infiltrating ab ILTCKs, but not in CD8^+^PD1^+^ T cells (Figure 4) [212,218].

Notably, these cells represent a distinct entity from the classical cytotoxic T cells. ILTCKs differ from traditional cytotoxic T cells by their unusual ontogeny, unique cancer cell-sensing mechanism, and antigen receptor self-reactivity. This reveals a novel subset of the cytotoxic T-cell population facilitating tumor immune surveillance. The ILTCKs are produced from specific thymic progenitors succeeding the initial interaction with cognate antigens and were constantly replenished by thymic progenitors during tumor growth. They generally react to the unmutated self-antigens [218]. IL-15 produced by cancer cells is a major driver for expansion and effector differentiation of intertumoral ILTCKs (Figure 4). Chou and his group demonstrated that deleting IL-15 from cancer cells eliminates the killer innate-like T cell protection and induces tumor growth [213]. Interestingly, these cells can be generated ex vivo and utilized in vivo to enhance the anti-tumor activity via inducible activation of IL-15 signaling. These αβILTCKs can identify tumor cells with a low mutational burden, and their mechanism of action is specific from the conventional PD-1^+^ CD8^+^ CTLs. Therefore, utilizing these αβILTCKs as a therapeutic target may be useful against tumors refractory to current checkpoint inhibitor therapies [218]. Intrinsically, ILTCKs could have been a novel target for small molecules to develop a new class of advanced cancer immunotherapy.

Several small molecules have been recognized for their capacity to elicit or modulate IL-15 production in αβ T cells. These encompass a range of small molecules (i) IL-15-based therapy and super-agonist: Investigations into IL-15-focused therapies, both independently and in conjunction with biological agents, monoclonal antibodies, and adoptive immunotherapy, have been documented [219]. The inherent short half-life of soluble IL-15 in vivo, an IL-15 super-agonist receptor-linker-IL-15 (RLI) has been engineered, incorporating soluble IL-15R components to enhance IL-15 bioavailability and bolster effector cell cytotoxicity. IL-15SA is currently under scrutiny as a potential cancer treatment or an enhancer of anti-tumor activity when combined with other interventions [220]. (ii) Short-chain fatty acids (SCFAs) metabolites generated by gut bacteria during dietary fiber fermentation, SCFAs like butyrate have shown potential in modulating immune cell functions. While some SCFAs, including butyrate, are associated with immune regulation and IL-15 expression, they can also impact IL-15 production [221]. (iii) Naturally occurring organic compounds, polyamines are implicated in cell growth and differentiation and have been linked to IL-15 expression in T cells [222], and (iv) Agonists of Toll-like receptors (TLR2 and TLR4) exhibit potential immunomodulatory effects, stimulating IL-15 induction and influencing immune responses [223]. These discoveries emphasize the complex spectrum of IL-15 modulation by various molecular agents, illuminating promising pathways for the therapeutic manipulation of ILT cells to enhance their robust anti-tumor response.

In breast, prostate, and colorectal cancer (CRC), (ILTCKs) have been identified, driven by tumor-derived IL-15. ILTCKs exhibit innate lymphocyte markers and robust cytotoxicity. Expressing αβ or γδ T cell receptors (TCRs) along with CD8α, ILTCKs possess an array of NK cell receptors (NK1.1, Ly49s, NKG2A/C/E, NKG2D, 2B4), and the signaling adapter Tyrobp, while lacking CD4, CD28, and ICOS co-stimulatory molecules. ILTCKs’ cytotoxic potential is evident via GzmB, GzmC, Perforin, and TRAIL expression. GzmB remains high in ILTCKs during late-stage tumor growth, suggesting resistance to exhaustion [214]. Parabiosis investigations unveil ILTCKs’ partial tissue localization, correlated with CD49a and CD103 expression, and trivial expression of trafficking receptors such as S1PR1, CCR2, CXCR4, and CX3CR1 [212,214]. These findings illuminate ILTCKs as a distinct and potent component of the tumor immune landscape, with implications for novel therapeutic strategies.

Another example of cell transformation is oncogene-induced tissue-resident lymphocyte response in murine cancer models. A study by Dadi et al. demonstrated that cell transformation induces the expansion of tissue-resident Innate Lymphoid Cells (ILCs) and Innate-like T Cells (ILTCs) with potent cytolytic activities against tumor cells [214]. These non-circulating cytotoxic lymphocytes are generated from innate T cell receptor (TCR)αβ and TCRγδ lineages, spread in primary tumors, and identified by high expression of NK1.1, integrins CD49a, and CD103 (Figure 4). Their specific gene-expression signature differs from conventional NK cells, T cells, and iNKT cells. The tumor-associated ILC1 is transcriptionally distinct but functionally related to cytotoxic NK cells. At the same time, these cells are transcriptionally similar and phenotypically different from the conventional CD127^+^ ILC1s. The CD4-CD8α-and CD4-CD8α+ subsets of TCR+ ILTC1s can be distinguished by examining the coreceptor’s expression. The differential expression of coreceptors along with high expression of CD103 and several NK cell receptors in TCRαβ+ ILTC1s was reminiscent of TCRαβ+ intestinal epithelial lymphocytes (IELs) [214,224]. ILC1s and ILTC1s production is dependent on the cytokine IL-15, just like αβILTCKs. It is yet uncertain, though, whether IL-15 encourages their activation in altered tissues, differentiation, and homeostasis. Killer innate-like T cells can detect potentially harmful antigens on their own without the aid of antigen-presenting cells like dendritic cells. They act more like innate immune cells, which are constantly prepared for an attack [225]. To sum up, ILTCKs stand out due to their distinct gene signatures, autoantigen reactivity, and broad tumor cell sensing capabilities, setting them apart from other intra-tumoral cytotoxic T cells and reported innate lymphoid cell lineages. This finding paves the way for novel therapeutic strategies, harnessing small molecules to expand ILTCKs within tumors, especially amidst anti-PD1 resistance. Further, research is essential to unravel the precise contributions of ILCs, ILTCs, and ILTCKs in shaping the TME and driving cancer progression. Exploring the potential of small molecules to target ILCs and ILTCs could offer opportunities to enhance cytotoxic activities. In addition, identifying the small molecules that potentiate these new immune “Soldier” expansion may provide an excellent asset in anti-cancer immunity. These strategies may lead to the development of novel immunotherapy for cancer treatments. Investigating epigenetic modifiers for ILTCK-related genes holds promise for augmenting ILTCK expansion and function. Manipulating metabolic pathways impacting immune cell behavior might amplify ILTCK responses. Moreover, the potential of ILTCKs in cell-based immunotherapies like CAR T cell therapy deserves attention. Continued efforts are required to delineate ILTCK roles within human tumors and their implications for ongoing immune therapies. As research progresses, these avenues may unveil innovative strategies for precise TME modulation and effective cancer treatment.

## 5. Conclusions

Traditional cancer treatments, including targeted immunotherapies, face persistent challenges that impact their effectiveness in treating cancer patients. To enhance treatment outcomes, it’s crucial to modulate the TME and reestablish immune surveillance by targeting specific components. An innovative approach in cancer therapy involves the use of small molecules to influence various immune cell populations within the TME, including both conventional tumor-infiltrating T cells and nonconventional ILT. These small molecules present a versatile strategy for fine-tuning immune responses within the TME, ultimately empowering the immune system to better detect and eliminate cancer cells. Through the precise targeting of specific signaling pathways and interactions, these compounds have the potential to overcome cancer’s immune evasion mechanisms, fostering a robust and efficient antitumor response. Furthermore, the exploration and combination of novel small molecules to target emerging immune targets may further augment the outcomes of cancer treatment. As research in this dynamic field progresses, the utilization and advancement of small molecules emerge as a promising prospect in the realm of cancer immunotherapy. This paves the way for innovative avenues in therapeutic intervention and has the potential for improved patient outcomes.

## Figures and Tables

**Figure 1 biomedicines-11-02621-f001:**
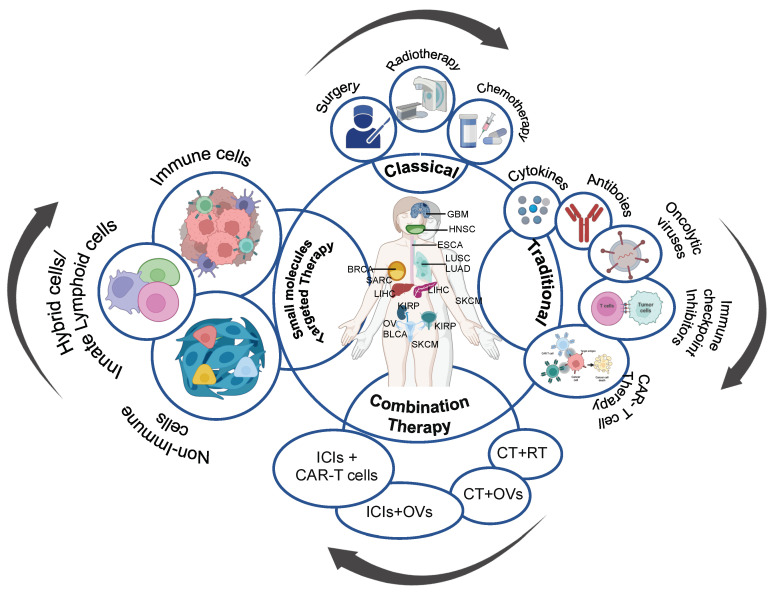
An overview of evolution and approaches of cancer therapy. This illustration represents classical, traditional, and innovative combination and small molecules targeted approach to treat cancer, combining different treatment modalities to acquire the most efficient and personalized therapy for patients. CT: Chemotherapy, RT: Radiotherapy, OVs: Oncolytic Viruses Therapy, and ICIs: Immune Checkpoint Inhibitors. ILCs: Innate Lymphoid Cells.

**Figure 2 biomedicines-11-02621-f002:**
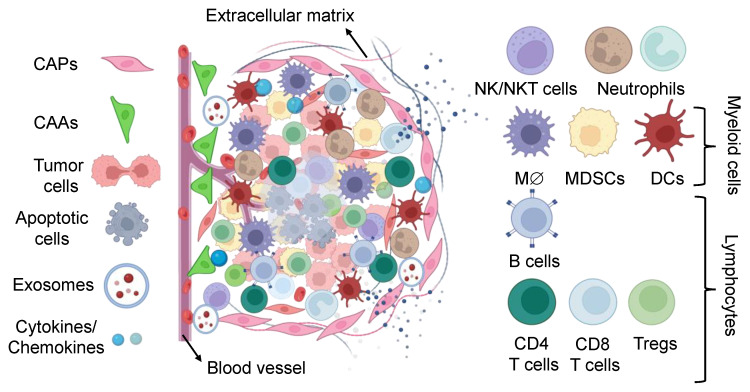
The tumor compositions. A schematic diagram illustrating the non-immune and immune components of the tumor microenvironment. Conventional tumors are fenced by innate and acquired immunity consisting of stromal and infiltrating immune cells, including lymphocytes, neutrophils, macrophages, MDSCs, dendritic cells, and NK cells. They create an intricate regulatory network that increases tumor proliferation by inducing tumor tolerance to evade immune surveillance and destruction. MDSC, myeloid-derived suppressive cells; TAM, tumor-associated macrophages; TAN, tumor-associated neutrophils; CAF, cancer-associated fibroblasts.

**Figure 3 biomedicines-11-02621-f003:**
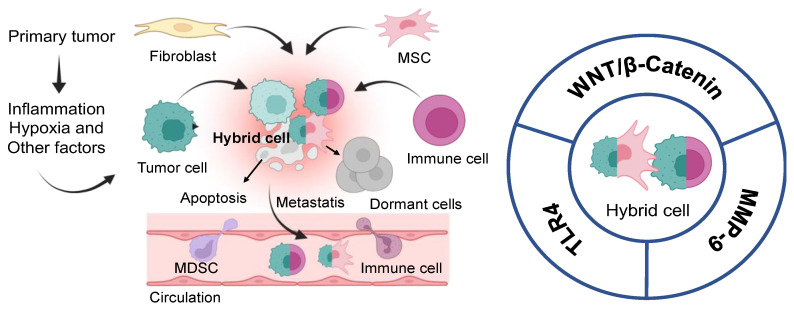
Illustration depicting the formation and fate of hybrid cells. Left panel: Several factors induce the fusion of tumor cells with fibroblast, mesenchymal stem cells, and immune cells. Most hybrid cells die and form metastases; some can enter a state of dormancy. Right panel: Tumor hybrid cells: potential therapeutic targets for small molecules. Wnt/β-catenin signaling pathway; MMP-9, matrix metalloproteinase 9; TLR4, Toll-like receptor 4.

**Figure 4 biomedicines-11-02621-f004:**
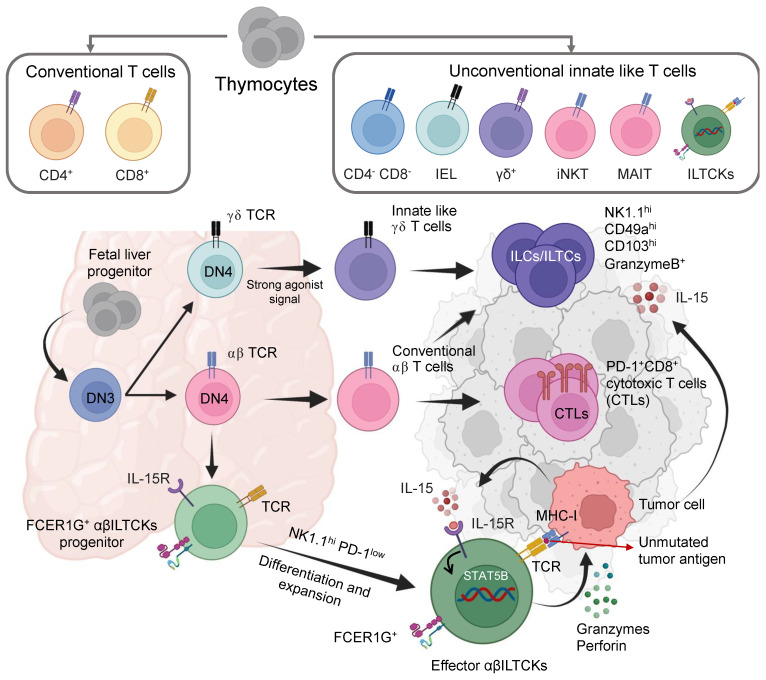
Differentiation, activation, and function of thymus-derived cytotoxic tissue-resident innate lymphoid cells, aka ”new immune soldier” in TME. Thymic progenitors give rise to both conventional (CD4^+^ and CD8^+^) and innate-like T cell subsets (CD4^–^CD8^–^, CD8αα IELs, γδ T cells, iNKT, MAIT, and CD8αα ILTCKs). The innate-like cells (ILCs) and innate-like T cells (ILTCs) are derived from innate T cell receptor TCRγδ (Top tier) and TCRαβ lineages (middle tier). Both are highly cytotoxic and are characterized by high expression of NK1.1, CD49a CD103, and GzmB production. Other innate-like T cell killers (αβILTCKs) are evolutionarily conserved innate T lymphocytes developed from a unique thymic progenitor with highly cytotoxic potential (GzmB, GzmC, Perforin, and TRAIL) and FCER1G expression. These FCER1G^+^ αβILTCKs are separated from conventional CTLs expressing high PD-1 responsive to unmutated self (tumor) antigen. IL-15 produced from the tumor cells promotes their intra-tumoral recruitment, local expansion, and effector activity. DN3: Doble negative three stages, DN4: Doble negative four stages, T cell receptor, MHC-I major histocompatibility complex class I, IL-15R IL-15 receptor, FCER1G Fc epsilon receptor Ig.

**Table 1 biomedicines-11-02621-t001:** Non-immune and immune cells play a critical role in TME for cancer progression.

TME Components Influence Tumor Progression		
**Non-Immune Cells**		
Type	Mediator	Key Role
Cancer-associated fibroblasts (CAFs)	α-Smooth muscle actin, fibroblast activation protein, vimentin, desmin, PDGFR α, and β	Support proliferation of tumor cells Angiogenesis Augment immune evasion, metastasis and drug-resistant
Tumor-associated adipocytes (TAPs)	Secretes adipokines, chemokines, cytokines, CCL2, CCL5, IL-1β, IL-6, TNF-α, VEGF and leptin	Promote tumor growth and control therapeutic response Carcinogenesis, tumor development and metastasis
Tumor-associated pericytes (TAPs)	Upregulate MHC class II in response to IFN-γ	TME immunomodulation, PD-L1 upregulation Cancer development and spreading
Tumor endothelial cells (TECs)	VWF, P-selectin (CD62P), and angiopoietin-2 (Angpt2)	Priming, activation, or down-regulation of effector immune cells Gatekeepers for immune cells that infiltrate TME
Extracellular vesicles (EVs)	IL-6, CXCL-1, CCL2, MDM2, α-SMA, VEGF, NKG2L and PD-L1	Promote tumor growth and angiogenic activity Impair NKG2D^+^ NK cell cytolytic activity Suppress PD-1^+^ T cell effector function Promote Treg cell expansion
**Immune cells**		
Subtype	Phenotype	Key role
Tumor-infiltrating Natural Killer cells (TINKs)	CD27^hi^, CD11b^hi^	MDSCs conversion to suppress CTLs Modulate ILCs Enhanced lymph node metastasis by promoting RORγt + ILC3 cells
Invariant NKT (iNKT)	CD1d restricted	Potent anti-tumor immunological response
Natural Killer T cells (NKT)	CD56, NKG2, CD94, CD161, NKG2D, NKG2A, NK1, Ly49	Inhibition, adhesion, activation, and recognition of cellular targets and healthy spare cells
Neutrophils and Tumor-associated neutrophils (TANs)	CD45, CD16^hi^, D11c, CXCR2^hi^, CXCR4^low^, CD62L^hi^, ICAM1, Arginase-1, TGF-β	Enhance and suppress tumor growth Suppress T cell activation Promote tumor growth
Macrophages	CD14, CD11c, CD16, CD64, CD68, CD206, HLA-DR, and CCR5	Generating cytokines, chemokines, and growth factors Facilitate tumor development and progression Suppress T-cell activity
Myeloid-derived suppressor cells (MDSCs)	CD14, CD15, HLA-DR, CD16, CD66b, CD11b and CD33	Immunosuppressive properties Inhibit anti-tumor immune response Stimulate tumorigenesis, tumor growth and metastasis
Dendritic cells (DCs)	ITGAX, CX3CR1, FLT3 and CSF1R	Stimulates anti-tumor T cell immunity Digestion of tumor-derived antigens and delivery
B-lymphocytes	Immature B cells (CD19, CD20, CD24, CD38, CD45R) Mature B cells (IgM and CD19)	Protumor effect: IL-10 and TGF-b Anti-tumor effects: activate NK cells and macrophages Antigen presentation and neutralization
CD4^+^ T helper cells	CD4^+^	Development and operation of CD8^+^ memory T cells
Cytotoxic T lymphocytes (CTLs)	CD8^+^	Anticancer immune response
Regulatory T cells (Tregs)	CD4^+^, CD25 and FOXP3	Immunosuppressive activity on CTL and helper CD4^+^ T cells Reduce antitumor immunity and promote tumor growth

**Table 2 biomedicines-11-02621-t002:** Small molecule inhibitors target TME components for tumor growth inhibition.

Small Molecule Inhibitor	TME Component	Cancer Type	Major Impact	References
Sorafenib	TAM	Breast cancer	Alter immunosuppressive cytokine profile of TAMs Restore IL-12 secretion Reduce IL-10production	[52]
Bindarit		Breast and prostate cancer	Decrease TAMs infiltration Spoil inflammatory cell responses Suppress MCP-1/CCL2synthesis	[53]
Trabectedin		Fibrosarcoma, ovarian and lung carcinoma	Cytotoxic activity on circulating monocytes and TAMs Deplete TAMs Triggers extrinsic TRAIL apoptotic pathway Suppress CCL2 and IL-6 production	[54,55]
Tasquinimod		Melanoma and prostate cancer	Suppress M2-polarized phenotype of TAM and MDSCs Targeting S100A9	[56]
BLZ-945		Breast and colon cancer	TAMs depletion Suppress CSF1 receptor	[57]
AS1517499		Breast	Macrophage differentiation Target STAT6 pathway Arginase1 (Arg-1) activity suppression Reduction in Mrc-1 and Arg-1	[58]
Paclitaxel	DC	Lung cancer	Suppress propagation of regDC Modulate Rho GTPase pathway	[59]
Sildenafil	MDSC	Multiple myeloma and head and neck cancer	Suppress MDSC function Activation and infiltration of intra-tumoral T-cells Impede phosphodiesterase-5 Downregulate expression of nitric oxide synthase-2 and Arg-1	[60]
Sunitinib		Metastatic renal cell carcinoma and pancreatic neuroendocrine tumor	Reduce Treg and MDSCs Prevent angiogenesis	[61,62]
GW2580		Prostate cancer	Reduce MDSC infiltration Inhibit CSF1/CSF1R signaling	[63]
Axitinib		Metastatic renal cell carcinoma	Reduced MDSC in spleen and tumor site Downregulate STAT-3 Alter MDSC-mediated tumor-induced immunosuppression	[64]
IC87114	TIL	Colon cancer and melanoma	Suppress Treg proliferation and activation Increase CD8þ response Suppress PI3K-Akt pathway	[65]
Trametinib		Colon cancer	Affect CD4þ T cell differentiation Reduce cytokine production Suppress MEK	[66]
SB415286		Melanoma and lymphoma	Activate CD8þ T cell function Downregulate PD-1 expression	[67]
RKN5755	CAF	Breast cancer	Suppress CAF migration Modulate b-arrestin-1 activity	[68]
WRG-28		Breast cancer	Reduced migration and invasion Allosteric interaction DDR2 suppression	[69]
Scriptaid		Melanoma	Decrease CAFs differentiation Reduce ECM secretion Induce HDAC inhibition	[70]
Navitoclax		Cholangiocarcinoma	Activate Treg apoptosis Upregulate Bax Reduce tenascin C expression	[71]
Sunitinib	EC	Colon, Renal cell carcinoma Epidermoid carcinoma	Arrest cell growth Target FLT3 receptor tyrosine kinase, PDGF, VEGF, and KIT	[72,73]
Dasatinib		Colon and prostate cancer	Inactivation of myeloid and ECs Decrease tumor growth Induce apoptosis Suppress SFKs phosphorylation MMP-9 reduction	[74]
TNP470		Breast cancer Glioblastoma Melanoma	Suppress vascular hyperpermeability Decrease VEGF phosphorylation Activation of RhoA and calcium influx	[75]
DIMP53-I		Colon cancer	Decrease tube formation and migration Suppress angiogenesis Inhibit p53 interaction with MDMX and MDM2 Induce apoptosis	[76]
Pazopanib		Metastatic renal cell carcinoma Multiple myeloma	Inhibit migration, growth, and survival Reduction in adhesion molecules	[73,77]
CC5079		Colon cancer	Decrease migration and proliferation of fibroblast and EC Suppress microvessel formation Trigger MKP1 expression	[78]
CX-4945		Pancreatic cancer Breast cancer	Induce cell cycle arrest Inhibit tube formation and migration Suppress PI3K/Akt pathway Block HIF-1α transcription	[79]
LLL12		Osteosarcoma	Decrease migration and proliferation Reduce tube formation Suppress phosphorylation of STAT	[80]
Biochanin A		Angiogenic glioma	Decrease viability, growth, migration, and invasion ERK/Akt/mTOR, VEGF, and HIF-1α inhibition	[81]
PD173074	CAF and EC	Head and neck squamous cell carcinoma	Decrease proliferation Suppress FGFR	[82]
Combination Study				
Cyclophosphamide + OX40	TIL	Melanoma	Induce Treg depletion Increase CD8þ T effector cells	[83]
TW-37 + radiotherapy	EC	Head and Neck cancer	Abolish sprouting and reduce proliferation Suppress BCl-2	[84]
BEZ235 + Verteporfin		Prostate cancer	Reduce proliferation Suppress PI3K activation	[85]
SOM230 + Gemcitabine	CAF	Pancreatic cancer	Activate sst1 receptor Suppress mTOR/4E-BP1 Decrease CAF protein synthesis	[86]
PT-100 + Oxaliplatin		Colon cancer	Suppress CAFs Decrease chemoresistance Targeting Fibroblast activation protein	[87]
AC1MMYR2 + Taxol		Breast cancer	Reduce invasion and migration Alter CAF reprogramming	[88]

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
