# Peer review of "Small Molecule Targeting Immune Cells: A Novel Approach for Cancer Treatment"

_biomedicines, 2023, doi:10.3390/biomedicines11102621_

Round 1

Reviewer 1 Report

This manuscript tries to focus on small molecules that target immune cells, and predominantly focuses on cancer therapy. Unfortunately the review is unfocused, and can't seem to focus. Discussion of antibodies and viruses and bacteria being used have nothing to do with small molecules, and it becomes frustrating to read all the paragraphs and text that has nothing to do with the title of the review. When the authors do discuss small molecules, they include many that kill cancer cells through non-immune mechanisms, like DNA alkylation. The authors try and add some discussion of new immune cells to target, but some simple google searching finds papers that discuss the results of studies in these areas, and these references seem to have been left out of the review. Overall I think the authors need to figure out what they want this review to be, as in its current state it is not very informative, and does not give much in the way of useful information. Perhaps if the title of the review was changed and the focus moved away from small molecules and into just cancer therapy via immune regulation it might make more sense. 

Much of the text needs serious editing, and the use of an editing service might be necessary to improve the review to the point where it is publishable. Several sections of the text also need revision, and again I'm not sure that the document is really salvageable without a serious re-write and refocusing. Below are some of the major issues I had reading it over: 

Page 2, line 50 the authors state "Cancer immunotherapies are antibodies that offer benefits in terms of efficacy and selectivity." but then list some small molecules as being included in this category. Many of the other categories are also not technically antibodies, maybe reword this section? Is a discussion of antibodies really necessary in a review about small molecules? No one categorizes antibodies as small molecules...

Page 3, lines 64-81, I'm not sure what the first 2 paragraphs on this page are trying to say, they seem like a number of disjointed ideas, maybe reorganize them so one paragraph details the successes in the field with current approaches and then the second lists the remaining challenges? 

Page 3, line 86, "Cytokines thereby induce proliferation and differentiation of the cells that modify the cellular functions." Are proliferation and differentiation not cellular functions? Not sure what this means.

Page 3, line 105, specific is spelled incorrectly (specific)

Page 4, line 139-146, I'd just delete this discussion, the older literature on microbes and live viruses is not particularly relevant anymore given how rapidly the field has developed over the last 20 years. Also, why is there this section on viruses and antibodies in a review about " Small molecules targeting immune cells"? Shouldn't this be focused on small molecules??

Page 6, line 219, Further is misspelled (Furter)

Page 12, line 275, the authors refer to "unpublished trials". This seems dangerous to make claims based on unpublished data that cannot be reviewed, and should probably be removed.

Page 12, line 280, CPB refers to "Checkpoint Blockade", not Checkpoint Blocked

Page 13, table 2, some of the structures are not reproduced well, and therefore appear to be missing bonds, specifically Orgovyx, Tepmetko, Pluvicto, Vonjo, and Krazati. Also, do all these molecules really belong in a review about immunotherapy for cancer? It's unclear whether these molecules kill tumor cells through immune-based mechanisms.

Section 5.1 lists a number of non-immune cell types that perhaps could be targeted for cancer therapy, but since the review is on " Small molecules targeting immune cells" I don't see why this section is included at all, it should probably be deleted.

Section 5.2 finally gets to some discussion about targeting immune cells, however the section is quite speculative, just listing immune cells that might be targeted for therapeutic purposes with small molecules. Frustratingly, many of these cells have already been targeted and the approaches listed have been reported to be explored. For example, there has been a recent report of small molecules enhancing NK cells killing tumor cells (Pharmaceutical Biology, 58:1, 357-366), and a number of CXCR2 antagonists have been reported (Cancer Letters 563 (2023) 216185). While there is some discussion of the use of small molecules in T-cell regulation, there appear to be some significant omissions to the review.

Moderate editing of English language required

Author Response

Rewier#1 Comments and Suggestions for Authors

This manuscript tries to focus on small molecules that target immune cells, and predominantly focuses on cancer therapy. Unfortunately the review is unfocused, and can't seem to focus. Discussion of antibodies and viruses and bacteria being used have nothing to do with small molecules, and it becomes frustrating to read all the paragraphs and text that has nothing to do with the title of the review.

Response: We appreciate reviewer’s concern and removed the section in the revised manuscript.

When the authors do discuss small molecules, they include many that kill cancer cells through non-immune mechanisms, like DNA alkylation. The authors try and add some discussion of new immune cells to target, but some simple google searching finds papers that discuss the results of studies in these areas, and these references seem to have been left out of the review.

Response: Thank you to the reviewer for bringing it to our attention. The discussion of small molecule targeting non-immune cells has been thoroughly edited. Small molecule targeting to the new immune cells has been elaborated to accommodate all new findings of the field. The sections have been extensively revised in the revised manuscript. Please see page: 12-14, lines: 171-236; page: 16, lines: 281-304; page: 20, lines: 367-373; page: 21-31, lines: 423-438; 467-494; 525-570; 581-631; 640-645; 652-677; 694-787; 808-811.

Overall I think the authors need to figure out what they want this review to be, as in its current state it is not very informative, and does not give much in the way of useful information. Perhaps if the title of the review was changed and the focus moved away from small molecules and into just cancer therapy via immune regulation it might make more sense. 

Response: We appreciate reviewer’s concern and rewrite many sections in the manuscript to keep focus to the subject. The abstract and conclusion has been revised.

Much of the text needs serious editing, and the use of an editing service might be necessary to improve the review to the point where it is publishable. Several sections of the text also need revision, and again I'm not sure that the document is really salvageable without a serious re-write and refocusing. Below are some of the major issues I had reading it over: 

Response: Thank you. The manuscript has been edited thoroughly, the typographical errors has been rectified, and many sections has been revised extensively to address reviewer’s concerns.

Page 2, line 50 the authors state "Cancer immunotherapies are antibodies that offer benefits in terms of efficacy and selectivity." but then list some small molecules as being included in this category. Many of the other categories are also not technically antibodies, maybe reword this section? Is a discussion of antibodies really necessary in a review about small molecules? No one categorizes antibodies as small molecules...

Response: The section has been removed in the revised manuscript.

Page 3, lines 64-81, I'm not sure what the first 2 paragraphs on this page are trying to say, they seem like a number of disjointed ideas, maybe reorganize them so one paragraph details the successes in the field with current approaches and then the second lists the remaining challenges? 

Response: We appreciate reviewer’s concern, and the section has been removed in the revised manuscript.

Page 3, line 86, "Cytokines thereby induce proliferation and differentiation of the cells that modify the cellular functions." Are proliferation and differentiation not cellular functions? Not sure what this means.

Response: The section has been removed in the revised manuscript.

Page 3, line 105, specific is spelled incorrectly (specific)

Response: The section has been removed in the revised manuscript.

Page 4, line 139-146, I'd just delete this discussion, the older literature on microbes and live viruses is not particularly relevant anymore given how rapidly the field has developed over the last 20 years. Also, why is there this section on viruses and antibodies in a review about " Small molecules targeting immune cells"? Shouldn't this be focused on small molecules??

Response: We appreciate reviewer’s concern, and the section has been removed in the revised manuscript.

Page 6, line 219, Further is misspelled (Furter)

Response: Typographical error has been rectified.

Page 12, line 275, the authors refer to "unpublished trials". This seems dangerous to make claims based on unpublished data that cannot be reviewed, and should probably be removed.

Response: We appreciate reviewer’s concern, and the section has been removed in the revised manuscript.

Page 12, line 280, CPB refers to "Checkpoint Blockade", not Checkpoint Blocked

Response: ‘Checkpoint Blocked’ has been replaced with ‘Checkpoint Blockade’ in the revised manuscript.

Page 13, table 2, some of the structures are not reproduced well, and therefore appear to be missing bonds, specifically Orgovyx, Tepmetko, Pluvicto, Vonjo, and Krazati. Also, do all these molecules really belong in a review about immunotherapy for cancer? It's unclear whether these molecules kill tumor cells through immune-based mechanisms.

Response: We appreciate reviewer’s concerns and redraw the structure of the small molecules using Chemdraw software. The small molecules represented in Table 2 are FDA approved compounds for cancer treatment. These small molecules targeting various candidates of essential pathways involved in different cancers. The section has been rewritten to fully address the concerns.  Please see page: 8, lines: 137-140.

Section 5.1 lists a number of non-immune cell types that perhaps could be targeted for cancer therapy, but since the review is on " Small molecules targeting immune cells" I don't see why this section is included at all, it should probably be deleted.

Response: We appreciate reviewer’s concern and rewrite the section to discuss the crucial role of non-immune cells influencing the behavior of immune cells in the tumor microenvironment that drives tumor progression. Targeting these cell types with small molecule restricts the effect of immune cells on tumor growth. Please see page: 12-13, lines: 171-204; page:14, lines: 209-237.

Section 5.2 finally gets to some discussion about targeting immune cells, however the section is quite speculative, just listing immune cells that might be targeted for therapeutic purposes with small molecules. Frustratingly, many of these cells have already been targeted and the approaches listed have been reported to be explored. For example, there has been a recent report of small molecules enhancing NK cells killing tumor cells (Pharmaceutical Biology, 58:1, 357-366), and a number of CXCR2 antagonists have been reported (Cancer Letters 563 (2023) 216185).

Response: Thank you. The discussion of the immune cells targeting small molecules has been revised with new specific information. Specifically, the recent report of small molecules enhancing NK cells killing tumor cells (see page: 16, lines: 298-304) and CXCR2 antagonists (see page: 20, lines: 367-373) has been discussed.  In addition, many other small molecules targeting immune cells for tumor regression has been added into the manuscript.

While there is some discussion of the use of small molecules in T-cell regulation, there appear to be some significant omissions to the review.

Response: We discussed the use of small molecules targeting CD4 (see page: 25-27, lines: 581-630), CD8 (see page: 27-29, lines: 640-645; 652-677; 694-715), TRM (see page: 29, lines: 718-751) and Treg (see page: 30-31, lines: 756-774; 780-787; 808-811; 827-829) cell types extensively as suggested.

Comments on the Quality of English Language

Moderate editing of English language required

Response: The English language in the manuscript has been edited thoroughly, the typographical errors has been rectified, and many sections has been revised extensively to address the concerns.

Reviewer 2 Report

Overall, the manuscript is thorough, detailed and informative.  It is dense at times. The figures are very helpful and clear.  There is a lot of value in the sections that review the immunomodulatory impacts of some traditional chemotherapies and some other medicines that are not traditionally considered tumor-directed therapies.  No obvious topics that were missed.  

This manuscript needs some review of the use of English language.  There are a large number of small issues to address.  The issues seem to become a little less frequent further into the manuscript.  Examples of some concerns are below:

Page 3, line 69--"relaps" should be "relapsed"; lines 74-75--sentence is missing a subject/noun; line 91--plural noun and singular verb; lines 89 and 97, use "first time" is in an awkward place in the sentence.

Page 5, typos in lines 152 and 162 "thead" and "rappidly"

Inventors column in Table 1 have a lot of errors in capitalization

Author Response

Reviewer#2 Comments and Suggestions for Authors

Overall, the manuscript is thorough, detailed and informative.  It is dense at times. The figures are very helpful and clear.  There is a lot of value in the sections that review the immunomodulatory impacts of some traditional chemotherapies and some other medicines that are not traditionally considered tumor-directed therapies.  No obvious topics that were missed.  

Comments on the Quality of English Language

This manuscript needs some review of the use of English language.  There are a large number of small issues to address.  The issues seem to become a little less frequent further into the manuscript.  Examples of some concerns are below:

Response: Thank you for the concerns. The English language in the manuscript has been edited thoroughly, the typographical errors has been rectified, and many section has been revised extensively to address the concerns.

Page 3, line 69--"relaps" should be "relapsed"; lines 74-75--sentence is missing a subject/noun; line 91--plural noun and singular verb; lines 89 and 97, use "first time" is in an awkward place in the sentence.

Response: Typographical and grammatical errors has been rectified. Please see page: 7, lines: 93;

Page 5, typos in lines 152 and 162 "thead" and "rappidly"

Response: Typographical errors has been rectified.

Inventors column in Table 1 have a lot of errors in capitalization

Response: The errors has been fixed.

Reviewer 3 Report

This subject of great complexity  should be better if you had tried to group the topics in several separated reviews !

Author Response

Reviewer#3 Comments and Suggestions for Authors

This subject of great complexity should be better if you had tried to group the topics in several separated reviews !

Response: Thank you. We appreciate reviewer’s concern, and many sections has been removed and the manuscript has been revised to keep focus into the subject of the review.

Round 2

Reviewer 1 Report

This review is improved somewhat by the changes the authors have made, but there are still a number of issues. The introduction is ~14 pages long, moves between many topics with no reasoning, and at then end of it I'm still not sure where things are going. Finally on page 15 we get to section 3.2 Targeting immune cells for cancer therapy, this seems like where the review should begin. I think if the authors can streamline the introduction significantly, it would make the review much more readable. Some other problems I found:

At the bottom of page 7, the statement "In addition, it has enhanced penetration and accessibility ability, and contributed to advancements in personalized medicine and precision oncology" is confusing, mostly because I'm not sure what "it" refers to. Also, it's unclear what " enhanced penetration and accessibility ability," means, does this refer to PK/PD or oral bioavailability?

The authors state at the bottom of page 7; "In 2001, FDA approved the first small molecule compound, “imatinib”, a tyrosine kinase inhibitor (TKI), for cancer treatment [11] " Many small molecules were approved before this for cancer treatment, they even show many of them. It is also unclear imatinib exhibits its effects by targeting an immune pathway, it binds to the kinase domain of bcr-abl, this is not usually listed as an immune pathway.

I'm not sure Table 2 belongs in the grant at all, why list the 89 molecules that are used for cancer therapy? Do any of these inhibit immune processes? Many will change immune responses, just do to their toxicity, but their listing doesn't add much to the review.

The authors then begin to discuss the tumor microenvironment, maybe the review would be better if it was focused on how molecules changed the TME? No introduction is given to this section, it's unclear why it's included.

Then there is a discussion and table which includes non immune cells and their role in the TME, I thought this was a review on small molecules in cancer immunotherapy? At this point I'm 13 pages into the text and we still are not engaged in the title topic.

After page 15 the review improves, and the discussion becomes much more focused. I'd suggest reorganizing the introduction and either adding some guiding text between sections to help the reader understand where the work is going or removing much of the 15 pages of introduction and coming up with an introduction that is a more appropriate length.

The english language still could use some work in places.

Author Response

Response to the reviewers’ comments

This review is improved somewhat by the changes the authors have made, but there are still a number of issues. The introduction is ~14 pages long, moves between many topics with no reasoning, and at then end of it I'm still not sure where things are going. Finally on page 15 we get to section 3.2 Targeting immune cells for cancer therapy, this seems like where the review should begin. I think if the authors can streamline the introduction significantly, it would make the review much more readable. Some other problems I found:

Response: We acknowledge the reviewer's concern and are confident that the introduction is suitably organized to convey the challenges associated with conventional immune therapies and emphasize the significance of targeting TME. In addition, the strategies of immunomodulatory small molecule inhibitors target to the immune cells in the TME for the cancer treatment. Furthermore, we have relocated the tables and condensed the introduction to a more concise four-page length.

At the bottom of page 7, the statement "In addition, it has enhanced penetration and accessibility ability, and contributed to advancements in personalized medicine and precision oncology" is confusing, mostly because I'm not sure what "it" refers to. Also, it's unclear what " enhanced penetration and accessibility ability," means, does this refer to PK/PD or oral bioavailability?

Response: We appreciate reviewer’s concern and rewrite the sentences to clearly represent the statement. “These small molecules show potential in penetrating the TME, enhancing immune cell accessibility to precisely target cancer or immune cells within the TME. This leads to increased potency and reduces non-specific cytotoxicity, thus advancing personalized medicine and precision oncology.” Please see page: 4, lines: 122-125.

The authors state at the bottom of page 7; "In 2001, FDA approved the first small molecule compound, “imatinib”, a tyrosine kinase inhibitor (TKI), for cancer treatment [11] " Many small molecules were approved before this for cancer treatment, they even show many of them. It is also unclear imatinib exhibits its effects by targeting an immune pathway, it binds to the kinase domain of bcr-abl, this is not usually listed as an immune pathway.

Response: We respectfully disagree with the reviewer and provide the evidence to support the statement in the manuscript. We added the references in the revised manuscript.

Many small molecules were approved before this for cancer treatment, they even show many of them.

In 2001, FDA approved the first small molecule compound, “imatinib”, a tyrosine kinase inhibitor (TKI), for clinical use in cancer treatment. (Imatinib mesylate--a new oral targeted therapy; N Engl J Med, 2002 Feb 28;346(9):683-93. PMID: 11870247; Signal Transduct Target Ther 2021 May 31;6(1):201. PMID: 34054126.

It is also unclear imatinib exhibits its effects by targeting an immune pathway, it binds to the kinase domain of bcr-abl, this is not usually listed as an immune pathway.

Recent studies have revealed that therapeutic doses of Imatinib impact diverse immune cell functions, including the differentiation of dendritic cells (DCs), impairing the proper function of T-cells and macrophages. (PMID: 17504122, PMID: 27030078). Please see page: 4, lines: 128-130.

I'm not sure Table 2 belongs in the grant at all, why list the 89 molecules that are used for cancer therapy? Do any of these inhibit immune processes? Many will change immune responses, just do to their toxicity, but their listing doesn't add much to the review.

 Response: We appreciate reviewer’s concern and move the Table 2 which represents 28 small molecules those recently approved by FDA as Supplementary Table 2.  These small molecules have direct or indirect impact on the immune response influencing the immune cell populations.

The authors then begin to discuss the tumor microenvironment, maybe the review would be better if it was focused on how molecules changed the TME? No introduction is given to this section, it's unclear why it's included.

 Response: We believe that the introduction contains concise statements that convey the idea, how small molecules can impact the components of the TME to enhance cancer treatment. Please see page: 3, lines: 99-109.

To clarify the reviewer’s comment: “Describing TME and exploring its potential as a target for small molecules is grounded in recognizing that the TME comprises all the key contributors to tumor progression and advancement. Effectively addressing these active participants within the TME promises to provide a genuine cure to hinder tumor advancement and progression. This is precisely why we delve into the TME's immune landscape, emphasizing the pivotal roles played by each immune element. This comprehensive understanding serves as the foundation for identifying potential small molecules capable of either enhancing the body's antitumor response or mitigating the inhibitory effects of immune elements within the TME. Our focus on the TME and small molecule interventions stems from the belief that this approach can potentially be a game-changer in the battle against cancer.”

Then there is a discussion and table which includes non immune cells and their role in the TME, I thought this was a review on small molecules in cancer immunotherapy? At this point I'm 13 pages into the text and we still are not engaged in the title topic.

Response: We politely disagree with the reviewer.  The non-immune cell section and the corresponding table discuss the crucial role of non-immune cells influencing the behavior of immune cells in the TME that drives tumor progression. Targeting these cell types with small molecules restrict the effect of immune cells on tumor growth. We believe this section is providing critical information for the development of cancer immunotherapy.

After page 15 the review improves, and the discussion becomes much more focused. I'd suggest reorganizing the introduction and either adding some guiding text between sections to help the reader understand where the work is going or removing much of the 15 pages of introduction and coming up with an introduction that is a more appropriate length.

Response: We appreciate reviewer’s concern and move the Tables from the introduction to make the section more concise, now spanning four pages, including Figure 1. We firmly believe that the introduction is suitably structured to effectively communicate both the challenges associated with conventional immune therapies and the significance of targeting the TME, along with the utilization of immunomodulatory small molecule inhibitors that target immune cells.
